

**Development of a chemical ionization mass spectrometry system for measurement of**
**atmospheric OH radical**
Wei Pu[1]*, Zhouxing Zou[1]*, Weihao Wang[1], David Tanner[2], Zhe Wang[3], and Tao Wang[1]
[1]Department of Civil and Environmental Engineering, The Hong Kong Polytechnic University,
Hong Kong, China
[2]School of Earth and Atmospheric Sciences, Georgia Institute of Technology, Atlanta, USA
[3]Division of Environment and Sustainability, The Hong Kong University of Science and
Technology, Hong Kong, China
Correspondence to: Tao Wang (cetwang@polyu.edu.hk)
*These authors contributed equally to this work





**Abstract.** The hydroxyl radical (OH) is the most important oxidant in the atmosphere and plays
a central role in tropospheric chemistry. Ambient OH is extremely difficult to measure because
of its low concentration and high reactivity. We have developed and optimized a chemical
ionization mass spectrometry (CIMS) system to measure OH based on ion-assisted mass
spectrometry. A calibration unit was developed based on chemical actinometry to convert
detected signals to OH concentration. Different types of ion sources ($^{210}$Po and corona source)
and scavenger gases (propane, $C_3F_6$, and $NO_2$) were compared. Radioactive ion source ($^{210}$Po
foils) was chosen for lower detection limits, and propane was selected for high elimination
efficiency and the negligible influence on the signal stability. The sensitivity of the CIMS
instrument to OH radicals is influenced by the efficiencies of titration reaction, ion conversion,
and ion transmission. Through adjusting their efficiencies by changing the flow rates and
voltages, optimal sensitivity was determined. The background noise from OH interferences
was reduced by adjusting the flow rate of scavenger gas. The CIMS system achieved a detection
limit of ~$0.15 \times 10^6$ molecules cm$^{-3}$ (signal/noise=2). The CIMS was then taken out to measure
ambient OH radicals at an urban site in Hong Kong in April 2019. An obvious diurnal pattern
of OH radicals was observed, with the highest concentration of ~$6\times10^6$ molecules cm$^{-3}$ at
midday and the lowest concentration of ~$0.25\times10^6$ molecules cm$^{-3}$ at night, with an overall
accuracy of about ±51%. The results demonstrated the capability of our CIMS for OH
measurements on clear days. The tests and results from our study provide a useful reference to
other researchers who wish to develop and apply the CIMS technique to measure OH and other
chemicals.




## 1. Introduction

The hydroxyl radical (OH) is the most important atmospheric cleansing agent and is responsible for the degradation and removal of most of trace gases (Crosley, 1997). In regions strongly affected by anthropogenic activities, reactions of OH with volatile organic compounds (VOCs) and carbon monoxide (CO) lead to the formation of organic peroxy ($RO_2$) and hydroperoxyl ($HO_2$) radicals. They react with NO to form nitrogen dioxide ($NO_2$), producing ozone ($O_3$) (e.g., Hofzumahaus et al., 2009). The reactions of OH with $NO_2$ and sulfur dioxide ($SO_2$) and the self- and cross-reactions of $RO_2$ and $HO_2$ transform the primary pollutants into low-vapor pressure gas molecules such as nitric acid ($HNO_3$), sulfuric acid ($H_2SO_4$), and highly oxidized organic molecules (HOMs) (Lu et al., 2012). In addition, the reaction with OH is the main removal pathway of methane, which is the third most important greenhouse gas. Therefore, OH plays key roles in major environmental issues such as photochemical pollution, acid rain, haze, and climate change (Kulmala et al., 2004; Wang et al., 2017; Calvert et al., 1985; Lu et al., 2019).

The importance of OH in tropospheric chemistry was first recognized by Levy (1971). Since then, concerted efforts have been made to develop techniques to measure OH in the atmosphere (Heard and Pilling, 2003). However, the low concentration, high reactivity, and short lifetime (<1 s) of OH make itself enormously difficult to be detected and quantified. The low concentration requires high sensitivities and small interferences in the instruments; the high reactivity demands a small loss in the ambient sampling, and the short lifetime requires measurement at a high temporal-spatial resolution. It is a huge challenge to meet all of these requirements by a measurement system (Lu et al., 2019).

During the past decades, three major techniques have been developed for in-situ OH measurements: differential optical absorption spectroscopy (DOAS) (Wennberg et al., 1990),



laser-induced fluorescence (LIF) (Perner et al., 1976), and chemical ionization mass
spectrometry (CIMS) (Eisele and Tanner, 1991). DOAS and LIF techniques directly measure
OH based on spectroscopic methods. The major advantage of DOAS is that it is self-calibrating
via the well-known Beer-Lambert law and thus does not require to separate a calibration device
(Heard and Pilling, 2003). DOAS often serves as a primary standard for comparisons with other
measurement techniques. However, the application of DOAS to the measurement of ambient
OH is limited due to the interferences from other atmospheric constituents (Heard and Pilling,
2003). LIF measures OH by using pulsed 308 nm single photon excitation of OH at low
pressure with temporally delayed detection of the resonant OH fluorescence (known as
fluorescence assay by gas expansion, FAGE) (Holland et al., 1995), and it requires calibration.
The LIF technique has the advantages of direct excitation of OH and good selectivity and
sensitivity (Heard and Pilling, 2003). Unlike DOAS and LIF, the CIMS technique measures
OH indirectly based on the ion-assisted mass spectrometry method. It employs a chemical
reaction scheme that OH is firstly titrated into $H_2SO_4$ and subsequently measured by a specific
chem-ionization method (Eisele and Tanner, 1991). CIMS has fewer interference and higher
sensitivity compared to either DOAS or LIF techniques for OH measurement because of the
higher collection efficiency of ions than photons (Heard and Pilling, 2003).  As a result, CIMS
processes the lowest detection limit for ambient OH measurement among the three techniques
(Heard and Pilling, 2003).
LIF has been the most widely used technique for OH measurement in laboratory and field
studies (Stone et al., 2012). However, some LIF instruments may suffer from interferences in
environments of rich VOCs and poor NOx.  Previous field measurements by LIF in forested
regions have observed OH concentrations that are three to five times higher than those
predicted by models with presently known OH sources and sinks (McKeen et al., 1997;
Lelieveld et al. 2008; Hofzumahaus et al., 2009; Whalley 2011; Lu et al. 2012; Mao et al. 2012;



Novelli et al. 2014; Feiner et al. 2016; Tan et al. 2017). The disagreements were first attributed
to the existence of an unknown source of OH in VOC-rich environment (Peeters et al., 2009;
Hofzumahaus et al., 2009), whereas later studies found positive artifacts in some LIF
instruments in such environments (Mao et al. 2012; Novelli et al. 2014; Feiner et al. 2016; Liu
et al. 2018). For example, Mao et al. (2012) attributed 40-60% observed OH signal at a
California forest to interference in their LIF by using a chemical method to remove the
interference. Recently, Liu et al. (2018) inferred the equivalent OH concentrations from
measurements of isoprene and its oxidation products over Amazon and found that the inferred
OH concentrations compared well with the simulated results. On the other hand, other groups
did not find evidence of the positive bias in their LIF systems, which have different design and
configurations, continued to attribute the model underestimated OH to the presence of OH
unknown source(s) at their study sites (Whalley 2011; Stone et al., 2012; Fuchs et al.2012; Lu
et al. 2012; Tan et al. 2017). It is highly desirable to deploy an alternative technique to re-
examine the OH issue in forested regions.
The CIMS technique for measuring OH was first developed at Georgia Institute of Technology
by Eisele and Tanner (1991). The system was further improved at National Center for
Atmospheric Research by reducing wall reactions (Eisele and Tanner., 1993), by reducing the
background signal (Tanner and Eisele., 1995) and by developing a better calibration system
(Tanner et al., 1997). Mauldin et al. (1998) modified the CIMS for measurement at an aircraft
platform during the First Aerosol Characterization Experiment (ACE1), and Edwards et al.
(2003) further upgraded the calibration system. Based on the design of Tanner et al. (1997),
another two CIMS instruments were developed at the Meteorological Observatory
Hohenpeissenberg, Germany, by Berresheim et al. (2000) and at the National University of
Ireland Galway by Berresheim et al. (2013). Kukui et al. (2008) developed a new version of





the CIMS instrument at the Centre National de la Recherche Scientifique (CNRS), France,
which allowed for OH measurements in a moderately polluted atmosphere.
We describe here a new CIMS system that has been tested and optimized at The Hong Kong
Polytechnic University (PolyU). The instrument was built at THS, Inc (Atlanta, Georgia) with
the same design as the CIMS from the group of Eisele and Tanner. The measurement principles,
configurations of the CIMS instrument, and a calibration unit are described in detail. Different
scavenger gases, ion sources, and primary ions detection was compared. In addition, the
sensitivity and noise of the CIMS instrument to OH radicals were tested by adjusting the flow
rates and voltages. Accordingly, their optimal settings were derived. Finally, the test result of
ambient OH measurement was presented. These results provide detailed technical information
for other researchers who wish to apply the CIMS for ambient OH measurement. To our
knowledge, this instrument is the first OH measuring CIMS in Asia.
**2. Measurement principles**
The measurement of hydroxyl radical (OH) in this study was made with a chemical ionization
mass spectrometry (CIMS) technique, which has been described previously (Tanner et al., 1997;
Sjostedt et al., 2007). Briefly, the ambient OH is titrated to $H_2SO_4$ by adding $SO_2$ into the
sample air flow, which initiates the following reaction sequence in the presence of oxygen and
water vapor:

19        $OH + SO_2 + M \rightarrow HSO_3 + M$          (R1)

20        $HSO_3 + O_2 \rightarrow SO_3 + HO_2$          (R2)

21        $SO_3 + 2H_2O \rightarrow H_2SO_4 + H_2O$         (R3)





However, $H_2SO_4$ in the atmosphere also contributes to the total $H_2SO_4$ concentration. To solve
the problem, a scavenger gas is periodically added into the sample air flow to remove OH
radicals. Then, $H_2SO_4$ produced from the reaction of OH and $SO_2$ can be obtained:
$$[H_2SO_4]_{OH} = [H_2SO_4]_{TS} - [H_2SO_4]_{BS} \qquad\qquad (E1)$$
$[H_2SO_4]_{TS}$ and $[H_2SO_4]_{BS}$ are $H_2SO_4$ concentrations with and without adding scavenger gas,
respectively.
Apart from the interference from the pre-existing $H_2SO_4$, the reactions of NO with peroxy
radicals ($HO_2+RO_2$), whose daytime concentrations are typically 1-2 orders of magnitude of
OH, can  produce OH in the sample flow (Sjostedt et al., 2007):
$$RO_2 + NO + O_2 \rightarrow R^{'}CHO + HO_2 + NO_2 \qquad\qquad (R4)$$
$$HO_2 + NO \rightarrow OH + NO_2 \qquad\qquad (R5)$$
Reaction 2 can also produce $HO_2$ radicals as intermediate products. To reduce the positive bias
from Reaction 5, another scavenger gas is added into the sample flow after $SO_2$ to scavenge
recycled OH radicals.
The $H_2SO_4$ is then converted into $HSO_4^-$ by chemical ionization in reaction with the $NO_3^-$
primary reactant ions:
$$H_2SO_4 + NO_3^- \cdot (HNO_3)_m \cdot (H_2O)_n \rightarrow HSO_4^- \cdot (HNO_3)_m(H_2O)_n + HNO_3 \qquad (R6)$$
$NO_3^- \cdot (HNO_3)_m \cdot (H_2O)_n$ are cluster ions of $NO_3^-$ reactant ions with neutral $HNO_3$ and/or $H_2O$
molecules, with m and n mostly of 0-2 and 0-3 (Berresheim et al., 2000). The $NO_3^- \cdot (HNO_3)_m \cdot$
$(H_2O)_n$ cluster ions are generated by the reaction of $HNO_3$ vapor with electrons (Fehsenfeld et
al., 1975):
$$HNO_3 + e^- \rightarrow NO_2^- + OH \qquad\qquad (R7)$$





$HNO_3 + NO_2^- \rightarrow NO_3^- + HONO$      (R8)
$NO_3^- + (HNO_3)_m + (H_2O)_n + M \rightarrow NO_3^- \cdot (HNO_3)_m \cdot (H_2O)_n + M$      (R9)
Where $e^-$ is emitted from an ion source. The OH radical (artificial OH) formed from primary
ion creation (Reaction 7) is not desirable and regards as noise signal, see details in Section
5.4.3. The $NO_3^- \cdot (HNO_3)_m \cdot (H_2O)_n$ and $HSO_4^- \cdot (HNO_3)_m (H_2O)_n$ are subsequently
dissociated by the collisional dissociation chamber (CDC):
$NO_3^- \cdot (HNO_3)_m \cdot (H_2O)_n + M \rightarrow NO_3^- + (HNO_3)_m + (H_2O)_n + M$      (R10)
$HSO_4^- \cdot (HNO_3)_m \cdot (H_2O)_n + M \rightarrow HSO_4^- + (HNO_3)_m + (H_2O)_n + M$      (R11)
The OH is finally detected by a mass spectrometer system as $HSO_4^-$ at 97 m/z.
**3. CIMS system**
Figure 1 shows the schematic of our CIMS system, which consists of two parts including a
sample inlet system and a mass spectrometer system. The sample inlet system has two regions:
chemical titration region and chemical ionization region. The chemical titration region is where
$H_2SO_4$ formed by the titration reaction of OH and $SO_2$. Chemical ionization region is for
converting $H_2SO_4$ into $HSO_4^-$ ion cluster. The mass spectrometer system consists of three parts
including a collisional dissociation chamber (CDC), an ion guide chamber (IGC), and an ion
detection chamber (IDC). $HSO_4^-$ ion cluster is dissociated to $HSO_4^-$ in the CDC, then refocused
in the IGC and finally detected in the IDC.
**3.1. Sample inlet**
During OH measurements, sample air at ambient temperature and pressure is first drawn into
a 5 cm diameter, 32 cm long stainless-steel tube by a blower. The flow velocity is measured
manually using a pitot. A scoop is attached to the front of the stainless-steel tube for turbulence
reduction. The central part of the flow is then drawn through a 1.6 cm diameter stainless steel



inlet into the chemical titration region with the flow rate being determined by a mass flow
controller (MKS, MFC company). The excess flow is vented back into the atmosphere via an
inlet blower.
**3.1.1 Chemical titration region**
The chemical titration region in Figure 1 is equipped with two pairs of opposed stainless steel
needle injectors. The first (front injectors) pair is installed at a 69 mm distance from the tube.
The distances between the first and second (rear injectors) pairs are 25.8 mm. To measure OH
radicals, $SO_2$ is continuously added into the sample flow from the front injectors to titrate OH
into $H_2SO_4$ (Reactions 1-3). In this study, we used $^{32}SO_2$ to titrate OH, and the purity of $SO_2$ is
0.9 vol.%.
As discussed above, atmospheric $H_2SO_4$ can contribute background signals for OH
measurements. Therefore, another flow is added through a zero-dead space four-way
electrically operated valve, which is automatically switched the injection positions of
scavenger gas and pure $N_2$ every 3 minutes (see the pulsed flow in Figure 1). When the
scavenger gas is added through the front injectors to the sample flow, $N_2$ is switched through
the rear injectors. CIMS is then running in background mode. Under this condition,
atmospheric OH simultaneously reacts with $SO_2$ and the scavenger gas, with the reaction of
OH with scavenger gas being much faster than $SO_2$. This configuration produces background
signal (BS) from the interferences of atmospheric $H_2SO_4$ and the ion source, with negligible
contribution from atmospheric OH. When the scavenger gas and $N_2$ are switched into the
sample flow through the rear and front injectors, respectively, CIMS is running in the signal
mode. Atmospheric OH is all titrated by $SO_2$ and the total signal (TS) is produced. In addition,
another flow of scavenger gas is added continuously into the sample flow through the rear
injectors to scavenge OH radicals generated from Reaction 5. The OH concentration is obtained





from the ratio of the difference between the total signal and the background signal to the
primary ion ($NO_3^-$) signal. (Tanner and Eisele, 1995):

$$[OH] = \frac{1}{C} \times \frac{\{HSO_4^-\}_{TS} - \{HSO_4^-\}_{BS}}{\{NO_3^-\}} \qquad\qquad (E2)$$

Where square brackets and text braces are used to denote concentrations and signal counts,
respectively. C is the calibration factor.
**3.1.2 Chemical ionization region**
The sample flow through the chemical titration region is then drawn into the chemical
ionization region and mixed with the sheath gas (Figure 1). The sheath gas flow is continuously
drawn into the chemical ionization region through the annular space between the 3.5 cm o.d.
and 1.2 cm o.d. stainless steel tubes by a diaphragm pump (KNF-813). These tubes are
concentric with the downstream end of the chemical titration region. The sheath gas is produced
by a zero-air generator (Thermo Electron Corporation, Model 111) attached with active
charcoal and silica gel. Therefore, particles, $SO_2$, $NO_x$ and other trace gases are removed
effectively from the sheath gas. Before entering the ionization region, $HNO_3$ vapor and the
scavenger gas are added continuously to the sheath gas. The $HNO_3$ vapor is obtained by $N_2$
carrier gas flow passing through the headspace of a reservoir of concentrated liquid $HNO_3$.
When $HNO_3$ doped sheath gas passes through the ion source (Figure 1), $NO_3^- \cdot (HNO_3)_m \cdot$
$(H_2O)_n$ reactant ions are produced by the reaction of $HNO_3$ and electrons (Reactions 7-9).
The $NO_3^- \cdot (HNO_3)_m \cdot (H_2O)_n$ reactant ions from the sheath gas then react with $H_2SO_4$
molecules from the sample air to form $HSO_4^- \cdot (HNO_3)_m (H_2O)_n$ cluster ions in the chemical
ionization region according to Reaction 6. Voltages are added on the sample and sheath flow
tubes to produce an electrical field to force the $NO_3^- \cdot (HNO_3)_m \cdot (H_2O)_n$ reactant ions to the
center of the chemical ionization region and enhance the interaction of reactant ions with





H$_2$SO$_4$. This effectively increases the signal levels and improves the sensitivity for OH
measurements.
The total flow (Figure 1) is then exhausted at the end of the chemical ionization region through
diaphragm pumps (Thomas, SK-668) and controlled by an MFC. To prevent the HNO$_3$ vapor
from corroding the pump and MFC and polluting the ambient air, the exhaust flow is first
filtered through active charcoal cartridges and then vented back into the atmosphere at a
distance of >10 m from the sampling point. A small portion of the total flow is finally drawn
into the mass spectrometer system through a 101.6 um diameter pinhole. The air molecules,
especially H$_2$O molecules may form higher-order clusters upon adiabatic expansion and cool
in the vacuum mass spectrometer system (Berresheim et al., 2000). To reduce this influence, a
small counterflow of N$_2$ buffer gas is added on the atmospheric pressure side of the pinhole
(Figure 1). And voltages are added at the positions of N$_2$ buffer and pinhole to force the ions
into the mass spectrometer system.
**3.2. Mass spectrometer system**
The mass spectrometer system is separated into three differentially pumped chambers with two
adjacent chambers being connected through a 4 mm pinhole (Figure 1). The first chamber
behind the pinhole is a collisional dissociation chamber (CDC). The pressure of the CDC is
typically maintained at around 0.5 hPa through a drag pump (Adixen, MDP 5011) and a scroll
pump (Agilent Technologies, IPD-3). The CDC has a high ion kinetic energy (i.e. high electric
field to number density ratio), and most of the entered cluster ions (e.g. $HSO_4^- \cdot$
$(HNO_3)_m(HO_2)_n$ and $NO_3^- \cdot (HNO_3)_m \cdot (HO_2)_n$ ) are dissociated in the CDC through
Reactions 10-11.
The second chamber is an octopole ion guide high vacuum chamber (IGC). In this chamber,
the pressure is maintained at about $1.3 \times 10^{-3}$ hPa through a turbo molecular pump (Agilent



Technologies, TwisTorr 304 Fs) and the same scroll pump. Here, the ions from the CDC are
refocused by an octopole ion guide and transported to the third chamber.
The third chamber (IDC) contains a quadrupole mass filter and detector with attached signal
amplifier electronics. The mass-selected ions of the quadrupole are amplified and detected by
a channeltron ion multiplier, and then counted based on standard techniques (Sjostedt et al.,
2007). This chamber maintains a pressure of about $2.6 \times 10^{-5}$ hPa through another turbo
molecular pump (Agilent Technologies, TwisTorr 304 Fs) and the same scroll pump.
**4. Calibration**
**4.1 Calibration principle**
The calibration of CIMS (Figure 2a) is achieved by controlled concentrations of OH radicals,
which is produced through photolysis of water vapor by 184.9 nm light (Tanner and Eisele,

12    1995):

13         $H_2O + h\nu$ (184.9 nm) $\rightarrow OH + H$          (R12)

14         $H + O_2 + M \rightarrow HO_2 + M$          (R13)

The calibration factor C is then determined based on the produced OH concentrations and
detected signals of $HSO_4^-$ and $NO_3^-$ according to Equation 2.
**4.2 Calibration unit**
Figure 2 shows the main components of the calibration unit. The length of the cuboid stainless
steel tube is 52 cm with 1.6 cm side length. A high-precision capacitance humidity
measurement hygrometer (Vaisala, HMP100) is connected at the front of the tube and is used
to measure the temperature $T$ and dew point temperature $T_d$ of the mixed air. The optical
elements for illumination are mounted at the end of the tubes (air outlet side) to minimize the





influence of wall loss during calibration. The optical elements consist of a Pen Ray mercury
lamp (Analytik Jena, UVP Pen Ray) and a bandpass filter. The bandpass filter blocks most of
the photons emitted from the Hg lamp except those at 184.9 nm.  A small $N_2$ flow is purged to
prevent UV absorption and the formation of ozone (Kukui et al., 2008). Finally, the transmitted
light enters the tube and photolyzes water vapor to produce OH radicals according to Reaction
12. The mixing ratio of water vapor in the air flow is controlled through the mix of the dry
synthetic air and humidity air from a water bubbler.
**4.3 Calibration quantification**
**4.3.1 OH quantification**
The concentrations of OH radicals produced from the water vapor photolysis reaction can be
described as follow:

12       $$[OH] = I \times t \times \sigma_{H_2O} \times \phi_{H_2O} \times [H_2O] \tag{E3}$$

Where $I$ and $t$ are the photon intensity (unit: photons $s^{-1}$ $cm^{-2}$) and the reaction time of $H_2O$
photolysis, respectively. [OH] and $[H_2O]$ are the concentrations of OH radicals and water vapor,
respectively, $\sigma_{H_2O}$ is the photolysis cross-section of water vapor at 184.9 nm ($7.22 \times 10^{-20}$ $cm^2$,
Cantrell et al., 1997 ) and $\phi_{H_2O}$ represents the photolysis quantum yield, which is assumed to
be 1.0 at 184.9 nm. $[H_2O]$ is calculated according to the temperature (T), saturated water vapor
pressure ($P_{H_2O}^{\circ}$) and relative humidity (RH) of the mixed air flow:

19       $$[H_2O] \quad = \quad \frac{RH \times P_{H_2O}^{\circ} \times 100}{RT} \times NA \tag{E4}$$

R is the ideal law constant and NA is the Avogadro's number. $I$ and $t$ are obtained as a combined
product $It$.
**4.3.2 *It* quantification**





The combined product $It$ is obtained based on the chemical actinometry method (Figure 2b).
This method measures $NO_x$ generated from $N_2O$ photolysis with the same calibration unit under
the same condition of the CIMS calibration. Since $N_2O$ photolysis generates $NO_x$ through the
illumination of the UV light with the same photon intensity as the $H_2O$ photolysis, product $It$
can be determined by measured $NO_x$ and $N_2O$ mixing ratios (Edwards et al., 2003).
Briefly, high purity $N_2O$ (99.9%) mixed with dry $N_2$ or dry synthetic gas flows into the
calibration unit. The photolysis of $N_2O$ leads to the formation of $NO_x$ in the presence of oxygen
and nitrogen through the following reactions (Edwards et al., 2003):

9            $N_2O + hv \ (184.9 \ nm) \rightarrow N_2 + O(\,^1D)$             (R14)

10           $O(\,^1D) + O_2 \rightarrow O(\,^3P) + O_2$             (R15)

11           $O(\,^1D) + N_2 \rightarrow O(\,^3P) + N_2$             (R16)

12           $O(\,^3P) + O_2 + M \rightarrow O_3 + M$             (R17)

13           $O(\,^1D) + N_2O \rightarrow 2NO$             (R18)

14           $O(\,^1D) + N_2O \rightarrow N_2 + O_2$             (R19)

The $O_3$ produced from Reaction 17 could oxidize NO to $NO_2$. Therefore, the photolysis of $N_2O$
eventually converts it to $NO_x$ which is concurrently measured by a commercial $NO_x$ detector
(Thermo, Model 42i-TL) by converting $NO_2$ to NO with a blue light converter and NO being
measured by the chemiluminescence technique. The combined product $It$ is a function of the
mixing ratios of $N_2O$, $N_2$, $O_2$, and produced $NO_x$:

$$It = \frac{(K_{15} \times [O_2] + K_{16} \times [N_2] + (K_{18} + K_{19}) \times [N_2O]) \times [NO_X]}{2 \times K_{18} \times \sigma_{N_2O} \times \phi_{N_2O} \times [N_2O]^2} \quad \text{(E5)}$$





where $K_{15}$, $K_{16}$, $K_{18}$, $K_{19}$ are the rate constants of Reaction 15, 16, 18 and 19, respectively,
whose values can be found elsewhere (Kurten et al., 2012). $\sigma_{N_2O}$ is the absorption cross-section
of $N_2O$ ($1.43 \times 10^{-19}$ cm$^2$, Kurten et al., 2012) and $\phi_{N_2O}$ is the photolysis quantum yield which
is assumed to be 1.0 (Kurten et al., 2012).
Ideally, the $N_2O$ actinometry experiment should be conducted with the same flow rate as in
water vapor photolysis experiment such that the reaction time of these two photolysis
experiments can be the same. However, at the flow rate suitable for CIMS calibration (10 slpm),
the concentration of $NO_x$ produced from $N_2O$ photolysis is close to the detection limit of the
$NO_x$ detector. Hence, the $N_2O$ actinometry experiment was carried out at a lower flow rate (3
and 6 slpm) to increase reaction time for photolysis and then the $NO_x$ production. The *It* values
for $N_2O$ photolysis experiment $\left(It_{N_2O}\right)$ and water vapor photolysis experiment $\left(It_{H_2O}\right)$ have
the following relationship:
$$It_{H_2O} = \frac{FR_{N_2O} \times It_{N_2O}}{FR_{H_2O}} \qquad\qquad\qquad (E6)$$
where $FR_{N_2O}$ and $FR_{H_2O}$ represent the flow rate for the experiment of $N_2O$ photolysis and
water vapor photolysis, respectively. Based on this equation, $It_{H_2O}$ can be obtained by scaling
$It_{N_2O}$ with the ratio of $FR_{N_2O}$ and $FR_{H_2O}$. The relation in Equation 6 between the product *It*
and flow rate is validated in the next section.
**4.3.2 *It* determination**
Figure 3 shows the results of $N_2O$ actinometry experiment. Figure 3a shows the $NO_x$ produced
as the function of $N_2O$ mixing ratios from 10% to 15% at different flow rates ($FR_{N_2O} = 3$, 6,
and 10 slpm). Generally, increasing $N_2O$ led to more production of $NO_x$, and the lower flow
rate resulted in longer reaction time and the higher $NO_x$ concentrations. In figure 3b, the
product *It* respect to different flow rate was calculated according to E5 based on the produced





NO$_x$ and N$_2$O result in Figure 3a. The product $It$ linearly increased with the inverse of the flow
rate, which validates the linear dependency between product $It$ and inverse of the flow rate
shown by E6. This linear depend is consistent with a previous study (Kurten et al., 2012). In
addition, the product $It$ was mostly independent on N$_2$O mixing ratios with variation from 10%
to 15%. Figure 3c shows the flow rate scaled product $It$ as a function of different N$_2$O mixing
ratio. Based on the E6, the flow rate scaled product $It$ ($It_{H_2O}$) is calculated from $It_{N_2O}$ in Figure
3b multiplying the ratio of FR$_{N_2O}$ (3, 6, and 10 slpm, respectively) to FR$_{H_2O}$ (10 slpm). The
flow rate scaled product $It$ variate from 1.37 to 1.53 x 10$^{11}$ at different flow rates and N$_2$O
mixing ratio. As a result, the value of  $It_{H_2O}$ was derived as 1.46 x 10$^{11}$ photon cm$^{-1}$ from the
average of flow rate scaled product $It$.
**4.3.3 Calibration result**
Figure 4 shows an example of a typical procedure for determining the calibration factor. The
instrument signals were continuously measured by adjusting H$_2$O concentrations without
changing other parameters. The different OH concentrations were calculated according to
Equation 3. For each step, the signal intensities (in Hz) of HSO$_4^-$ and NO$_3^-$ were collected for 6
minutes with background mode and signal mode of each 3 minutes. The calibration factors
were determined from the calculated OH concentrations and signal intensities based on
Equation 2. The red dots in Figure 4 represent the average calibration factors for every 3
minutes. The result shows that the calibration factors for different steps were very close from
1.60 to 1.69x10$^{-10}$ and were independent of water vapor concentrations, which indicates the
high performance of calibration quantification. Then, the averaged calibration factor for our
CIMS is 1.64 x10$^{-10}$ molecule/cm$^{-3}$.
**5. Optimizations of instrument performance**



As shown in Figure 1, the CIMS system is complicated, and its performance is determined by
different parameters and components. In order to improve the performance of the CIMS for
OH measurement, different types of ion sources ($^{210}$Po radioactive ion source, corona source),
scavenger gases (propane, $C_3F_6$, and $NO_2$), and primary ions detection were compared.
Moreover, the instrument sensitivity and noise were optimized by adjusting the flow rates and
voltages.
**5.1. Ion source**
Radioactive ion source ($^{210}$Po or $^{241}$Am) and corona discharge source (corona ionizer) were
generally used as the ion source in previous studies (Berresheim et al., 2000; Sjostedt et al.,
2007; Kukui et al., 2008). In this study, $^{210}$Po and corona sources were compared.
**5.1.1 $^{210}$Po**
The $^{210}$Po acts as an ion source through the alpha decay. Briefly, $^{210}$Po emitted alpha particles
that interact with the carrier gas to quickly form thermalized electrons and positive ions
(Fehsenfeld et al., 1975). The formed electrons react with $O_2$ and then $HNO_3$ to produce $NO_3^-$ ·
$(HNO_3)_m \cdot (H_2O)_n$ reactant ions for ion conversion. Radioactive ion source such as $^{210}$Po was
usually used due to the low OH interference and convenience of installation. We compared
$^{210}$Po ions source and corona source (see below) for the performance on OH measurements
(Figure S1). The result showed that the detection limit by $^{210}$Po ions source was lower than the
corona source. In this study, $^{210}$Po foils were chosen as the ion source in our CIMS system. We
note that the radioactive ion source is usually subject to stringent health safety regulations and
the users should apply a permit to use the radioactive source. In addition, $^{210}$Po is an isotope of
polonium and undergoes alpha decay to stable $^{206}$Pb with a half-life of about 140 days.
Therefore, in order to keep stable signal intensities for primary ions, $^{210}$Po foils need to be
replaced regularly.



### 5.1.2 Corona source

Corona ionizer generates $NO_3^-$ by the discharge formed between the tungsten needle and a 1 mm diameter plate 3 mm from the needle tip (Kukui et al., 2008). Corona source has the advantage of producing much higher concentrations of $NO_3^- \cdot (HNO_3)_m \cdot (H_2O)_n$ primary ions compared with radioactive $^{210}Po$ or $^{241}Am$ foils, which leads to higher concentrations of $HSO_4^-$ and higher signal intensities (higher sensitivities). However, the corona discharge source is known to produce a significant amount of neutral species including OH radicals. When using the corona discharge source in our CIMS instrument, the concentrations of produced OH artifacts were much higher than those found in the ambient atmosphere (Figure S1). Although scavenger gas added in the sheath flow can remove most artificial OH radicals, the remaining interferences are still comparable with those in ambient environments and can deteriorate the detection limit of the CIMS instrument for OH measurements.

### 5.2 Scavenger gas

As discussed in Section 3, scavenger gas is very important for determining the performance of OH measurement by CIMS. Three types of scavenger gas (propane, $C_3F_6$, and $NO_2$) have been used in previous studies by different groups (Berresheim et al., 2000; Sjostedt et al., 2007; Kukui et al., 2008). However, there have been no reports on comparisons of these three scavenger gases. In this study, they were tested in the laboratory. A scavenger gas was added in two positions for different purposes. In the sheath flow, the scavenger gas reduced the interference of OH artifacts from the ion source; in the sample flow, it was injected to eliminate the ambient OH to determine background.

### 5.2.1 Propane

For propane, 99.95 vol.% pure propane (purchase from the Harvest Wise Gases (H.K.) Company) added in sheath flow could effectively (~80% see Figure 6) remove artificial OH



radicals from the ion source, and the remaining contributed a low and stable signal intensity at
97 m/z. For the OH removal efficiency of propane in sample air, even at OH concentrations of
two orders of magnitude higher than ambient OH level, propane could remove OH at 97.7%
(for more details, see Section 5.4.3). In addition, the signal intensity of the primary ions was
not affected by the added propane and kept stable (Figure S2). Propane is cheap and easy to
purchase. In our CIMS system, propane was selected as the scavenger gas.
**5.2.2 $C_3F_6$**
For $C_3F_6$, although its OH removal efficiency was high enough, it was found to suppress the
signal intensities detected by the mass spectrometer system. With no $C_3F_6$, the signal intensity
at 64 m/z for primary ions was quite stable even within one month. However, once $C_3F_6$ was
added, the signal intensity at 64 m/z declined quickly as shown in Figure S2. As a result, the
sensitivity for OH measurements decreased and the detection limit increased. Initially, we
suspected that the purchased $C_3F_6$ cylinder gas had high impurities, which may consume $NO_3^-$
ions. But, after replacing three different $C_3F_6$ cylinders gas which was purchased from two
different suppliers, the problem remained. We suspect that $C_3F_6$ may suppress the ion detection
efficiency of the mass spectrometer system.
**5.2.3 $NO_2$**
$NO_2$ was found not only to remove OH radicals but also to $HO_2$ radicals by converting them
into $HO_2NO_2$ when NO was present in sample air. This means that compared to the other
scavenger gases, $NO_2$ provides a better measurement accuracy, especially in a high NO
environment. However, only 1.0 vol.% $NO_2$ cylinder gas was available for our experiment due
to restrictions of using higher concentrations of $NO_2$, and its removal efficiency was not high
enough to remove OH. We note that high purity $NO_2$ gas is very dangerous and must be handled
with extreme caution.



**5.3 Primary ions detection**
Determination of the concentrations of OH radicals need to use the signal intensities of $NO_3^-$
ions according to Equation 2. In our study, $NO_3^-$ primary ions are detected by the mass
spectrometer system at 64 m/z. Some studies traced the $NO_3^-$ ions based on the signal intensities
at 62 m/z (Tanner et al., 1997). However, we noticed that the concentrations of $NO_3^-$ produced
in the inlet system are extremely high. Even though a small portion of the $NO_3^-$ ions are finally
detected by the mass spectrometer system, they can yield very strong signal intensities. After
operating the CIMS instrument with detecting the signal of $NO_3^-$ ions at 62 m/z about half a
year, we found a significant decrease in the signal intensity at 62 m/z (with all instrument
settings unchanged), which may be due to the accelerated aging of the channeltron detector by
the high $NO_3^-$ ions concentrations. Therefore, the isotopic signal ($N^{18}O_3^-$)at 64 m/z was chosen
to detect $NO_3^-$ primary ions for extended operation. The signal intensity at 64 m/z is lower than
at 62 m/z by about a factor of 167 (Kurten et al., 2012).
**5.4 Instrument sensitivity and noise**
**5.4.1 Parameters influencing the sensitivity**
The sensitivity (S) of the CIMS instrument to the OH radicals dependent on the reaction
efficiency of OH and $SO_2$ in chemical titration region (f(RE)), the conversion efficiency of
$H_2SO_4$ to $HSO_4^-$ in chemical ionization region (f(CE)), and the transmitted efficiency of $HSO_4^-$
from sample inlet to mass spectrometer system (f(TE)):
$$S \sim f(RE) \cdot f(CE) \cdot f(TE)$$
f(RE) is dependent on the reaction time and the reaction rate, which is mainly related to the
velocity of sample air flow and the concentration of $SO_2$. f(CE) is mainly controlled by the
mixing of $H_2SO_4$ in sample flow and $NO_3^- \cdot (HNO_3)_m \cdot (HO_2)_n$ primary ions in sheath flow in



the chemical ionization region, which is not only dependent on the mixing of sample flow and
sheath flow but also the voltages on the flow tubes as discussed in Section 3.1. f(RE) is related
to the $N_2$ buffer gas flow and voltages. We can optimize the sensitivity (S) by adjusting the
$SO_2$ flow, sample/sheath flows, $N_2$ buffer gas flow, and voltages.
**5.4.2 Sensitivity optimization**
**5.4.2.1 Optimization of $SO_2$ gas flow**
Figure 5a shows the normalized signal intensity (NSI) at 97 m/z for $HSO_4^-$ as a function of the
flow rate of $SO_2$ (0.9 vol.%). The NSI first increased with increased $SO_2$. Then, the NSI became
independent of the added $SO_2$ amount and a stable signal was obtained with flow rate > ~2.5
sccm. We set the $SO_2$ flow rate 5 sccm for a factor of 2 margin sccm for our operation,
following the previous study (Sjostedt et al., 2007).
**5.4.2.2 Optimization of sample/sheath flow**
Figure 5b shows the NSI as a function of the ratio of sample flow to sheath flow. The NSI
firstly increased and then decreased with the increased ratio, with a peak value at a
sample/sheath flow ratio of 0.3. This ratio was independent of the sample flow rates from 12
to 21 slpm. We think that this ratio produced a turbulent total flow in the chemical ionization
region, facilitating a fast mixing of the reactants, enhancing the ionization efficiency of $H_2SO_4$,
and increasing NSI at 97 m/z.    (Tanner and Eisele, 1995; Tanner et al., 1997).
The effects of the sample flow rate on NSI are shown in Figure 5c. The ratio of sample to
sheath flow was fixed as 0.3. Briefly, the NSI increased with the decreased flow rate. The result
is expected as the lower the sample flow rate is, the longer times Reactions 1-3, as well as
Reaction 6, have, which improves both f(RE) and f(CE). However, the increased reaction time
of Reactions 1-3 will increase the OH interference produced from the $HO_2$ recycling in the
presence of NO in sample air. Previous studies usually kept the reaction time less than 60 ms





(e.g. Tanner et al., 1997). After considering these two factors, the sample and sheath flow rates
were set at 3.7 and 12.6 slpm, respectively, which gives a reaction time of ~47 ms. The optimal
f(RE) is determined.
**5.4.2.3 Optimization of voltages**
Figure 5d-e shows the effects of voltages on NSI. As mentioned above, the voltages are applied
to force the ions to the centre of the chemical ionization region and enhances the ionization
efficiency. Similar to the flow ratio, the increase of voltage difference (inlet voltage minus
sheath voltage) firstly increased the NSI and then decreased it (Figure 5d). At the voltage
difference of 48 V, the peak NSI was achieved. In Figure 5e, the NSI increased with the
negative sheath voltage and then kept stable with sheath voltage < -70V (the voltage difference
was fixed as 48 V). based on these results, we set the inlet and sheath voltages at -32 and -80
V, respectively. At these settings, the optimal f(CE) was determined. The cross interactions of
sample/sheath flow and voltages on NSI were also evaluated in Figure S3. The highest NSI
was achieved when the sample/sheath flow ratio was close to 0.3 for different voltages. This
result indicates the response of NSI on voltages is relatively stable at the flow ratio of 0.3. The
voltages added to the pinhole also force the ion to the central of pinhole and pass through it
which are set at -70 and -40V.
**5.4.2.4 Optimization of $N_2$ buffer gas flow**
Generally, the mass flow into the mass spectrometer system is fixed and the $N_2$ buffer gas just
changes the amount of sample air versus dry $N_2$. Figure 5f shows the effect of $N_2$ buffer gas
flow. Generally, the NSI increased with the decreased flow rate since more ions in sample air
entered the mass spectrometer system and further increased the f(TE). However, a lower flow
rate of $N_2$ buffer gas also makes more neutral molecules in sample air entering the mass


spectrometer system and influencing the instrument (Berresheim et al., 2000). Therefore, the
flow rate of $N_2$ buffer gas was set as 440 sccm, and the optimal f(TE) was determined.
**5.4.3 Noise minimization**
As discussed above, the instrument noises for OH measurements are from $H_2SO_4$ in ambient
air and OH produced by the ion source. These noises can be reduced by adding a scavenger
gas.
In order to minimize the artificial OH produced by ion source, the scavenger gas was added to
sheath flow. Figure 6 shows the signal intensity at 97 m/z where $N_2$ gas was used as sample air
so that there were no OH radicals in sample air. The artificial OH signal from the $^{210}$Po ion
source was ~$3.5\times10^6$ molecules cm$^{-3}$ without propane in sheath gas, which is comparable to
the typical OH concentrations in ambient environments. When propane was added into the
sheath gas, the artificial signals were effectively reduced to less than ~$1\times10^6$ molecules cm$^{-3}$
and kept stable when the flow rate was higher than 1 sccm. Based on the result, we set a flow
rate of 2 sccm for propane in the sheath flow.
To quantify and subsequently remove the contribution of ambient $H_2SO_4$ to signal at 97 m/z,
the scavenger gas was switched to the front injector to eliminate all ambient OH for the
detection of ambient $H_2SO_4$ signal only. The OH removal efficiency (RE) of the scavenger gas
in sample flow can influence the instrument's performance. Figure 7 shows the RE of propane
added in the sample flow. Excessive OH radical compared to ambient levels were produced
from the OH calibration source. The RE increased with the increased propane flow rate initially
and started to level off when more propane was added. We adopt the flow rate of propane at 2
sccm in front injector (also in rear injector), which led to~98% removal efficiency for OH and
allowed quantification of ambient $H_2SO_4$ for subsequent noise subtraction (E2).
**6. Detection limit and uncertainty**



The detection limit can be calculated as follows,

$$DL = \frac{1}{C} \times \frac{n * \sigma}{\{NO_3^-\}}$$
(E7)

Where DL is the detection limit in $10^6$ molecule/cm$^3$, C is the calibration factor, and $n$ is the
ratio of signal to noise S/N. $\sigma$ represents the standard deviation of the signal intensity of $HSO_4^-$
at 97 m/z, and $\{NO_3^-\}$ represents the signal intensity of $NO_3^-$ at 64 m/z at the integration time t.
Figure 8 shows the concentrations of OH radicals and the corresponding detection limit (S/N=2,
average time=6 minutes) in the laboratory. The detection limit was quite stable over the whole
day and ranged from 0.08 to 0.20*$10^6$ molecule cm$^{-3}$, with an average value of approximately
0.15 *$10^6$ molecule cm$^{-3}$.
The uncertainty for the calibration factor (C) of OH measurements is dependent on the
uncertainties of all the parameters involved in the calculation of the concentrations of OH
radicals and the precision of the measurements of signal at 64 m/z and 97 m/z. The uncertainty
was ~36% for $It$ (see Figure 3), $\sigma_{H_2O}$~5% for $\sigma_{H_2O}$, <1% for $\phi_{H_2O}$ (Cantrell et al. 1997), and
~10% for the water concentration (Kukui et al., 2008). The precision of the measurements
signal at 64 m/z and 97 m/z of the CIMS instrument ($2\sigma$) was 11% (for 6 min integration time).
The overall uncertainty for the calibration factor was about 38%.
**7. Field deployment of CIMS**
In order to examine the performance of our CIMS in the ambient environment, we deployed
the optimized instrument to an urban site of Hong Kong in April 2019 (Figure S4). The site
was located on the 11$^{th}$ floor of a teaching building on the campus of The Hong Kong
Polytechnic University (PolyU) and was surrounded by several busy roads. The sample inlet
was positioned horizontally facing the south. Measurements were made with a time resolution
of 10 seconds. A typical measurement sequence consisted of 3 minutes in the background mode
and 3 minutes in the signal mode. Figure 9a shows the diurnal profile of OH concentrations (3-
minute average) observed on April 25, 2019, and the solar radiation measured using UTA-
LI200 at a time resolution of 1 minute. Figure 9b shows the measured signal intensities at 97
m/z at the signal mode and the background mode. The OH concentrations exhibited a clear
diurnal profile with the highest value of ~$6\times10^6$ molecules cm$^{-3}$ at midday and the lowest level
of ~$0.25\times10^6$ molecules cm$^{-3}$ at night. The OH concentrations were highly correlated to solar
radiation, which was similar to previous studies (e.g. Rohrer and Berresheim, 2006; Tan et al.,
2017). The 3-minute average OH concentrations were above the detection limits ($0.5\text{-}2\times10^6$
molecules cm$^{-3}$) most of the daytime, except during a cloudy period (08:00 to 10:00) (Figure
9a). This preliminary result demonstrated the capability of our CIMS for measuring ambient
OH on a clear day in an urban environment. However, Figure 9b reveals that the contribution
to instrument background from ambient $H_2SO_4$ was significant at the site, which raised the
detection limit and measurement uncertainty (to 51%). Future work will make use of
isotopically labelled $^{34}SO_2$ to eliminate $H_2SO_4$ interference.
**8. Summary and conclusions**
To measure the atmospheric OH radicals, we have developed the first chemical ionization mass
spectrometry (CIMS) system in Asia. It is an indirect measurement technique that converts OH
radicals to $HSO_4^-$ which is detected by the ion-assisted mass spectrometry method. In addition,
the calibration system has been developed. A series of comparisons of different ion sources,
scavenger gases, and primary ions detection have been conducted to optimize the performance
of the CIMS for OH measurement. The sensitivity is dependent on the efficiencies of titration
reaction, ion conversion, and ion transmission which have been improved by optimizing the
flow rates of a myriad of gases and voltages in various components. An initial field test has
demonstrated the capacity of this instrument in measuring ambient OH in an urban site on clear
days. The main findings on the key parameters are summarized below.





(1) $^{210}$Po has lower artificial OH interference compared to a corona ionizer, and it is

2          adopted as the ion source.

(2) $C_3H_8$ is a better OH scavenger than $C_3F_6$ and $NO_2$ (low concentration) because of the

4          high elimination efficiency and signal stability of $C_3H_8$.

(3) A procedure has been developed to optimize the flow rates of sample gas, sheath gas,

6          and $N_2$ buffer gas, voltages on the sample inlet system and the concentration of $SO_2$

7          titration gas with the aim to increase instrument's sensitivity and reduce noise.

(4) The CIMS instrument achieved a detection limit of $0.15 \times 10^6$ molecules $cm^{-3}$ and

9          uncertainty of 38% (S/N=2) under laboratory conditions. In the field, the detection

10          limit increased to about $0.15 \times 10^6$ molecules $cm^{-3}$ on clear days, with the overall

11          accuracy of about 51%.

(5) Future work includes more field experiments in various environments and utilization

13          of isotopically labelled $^{34}SO_2$ to eliminate the $H_2SO_4$ interference.

We note that the optimal values of instrument parameters may differ in different CIMS systems
due to the different design and/or configurations, the test procedures and results from our study
provide a useful reference to other researchers who wish to apply CIMS technique to measure
atmospheric OH radicals.



**Data availability.** All data used to produce this study can be obtained by contacting Tao Wang (cetwang@polyu.edu.hk)

**Supplement.** The supplement related to this article is available on line at:

**Author contribution.** TW invited the project, WP and ZZ designed and performed experiments with contributions from WW, ZW, DT and TW. WP, ZZ, and TW write the paper with contributions from DT and ZW.

**Competing interests.** The authors declare that they have no conflict of interest.

**Acknowledgments**

We thank the Environmental Protection Department of Hong Kong for loaning the CIMS instrument.

**Financial support.** This research was financially supported by the Hong Kong Research Grants Council (T24-504/17-N and A-PolyU502/16)

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





**Figures**

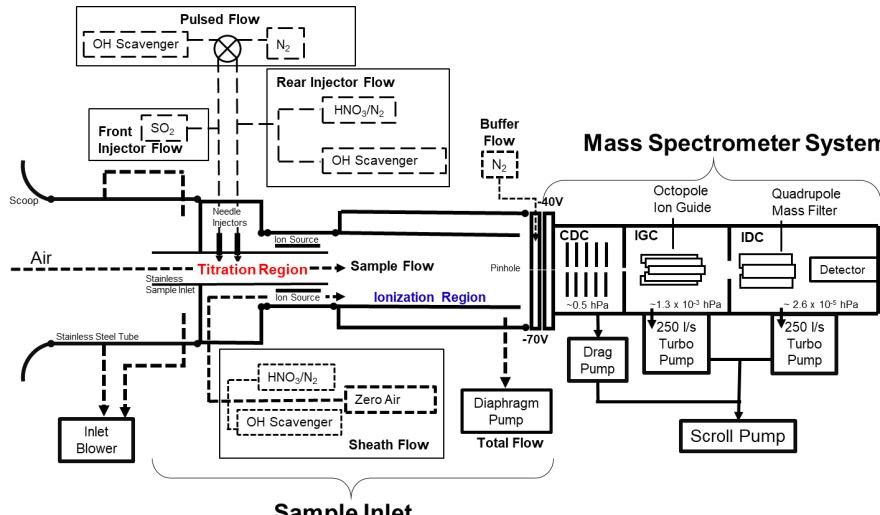

**Figure 1**. Schematic diagram of the OH-CIMS system, which including a sample inlet system and a
mass spectrometer system. The sample inlet system including chemical titration region and chemical
ionization region. The mass spectrometer system consists by a collisional dissociation chamber
(CDC), an ion guide chamber (IGC), and an ion detection chamber (IDC). Dashed lines depict the air
flows in the sample inlet during operations.

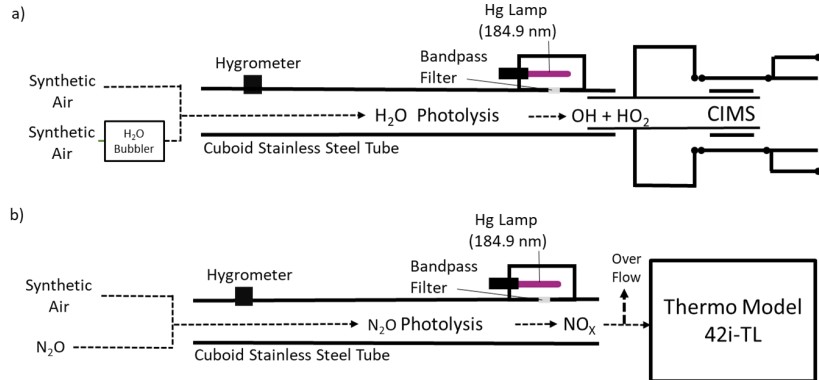

**Figure 2.** Schematic diagram of calibration. a) The CIMS calibration experiment. b) The $N_2O$
actinometry experiment for determination of product *It* value. Dashed lines show the air flows during
experiment.

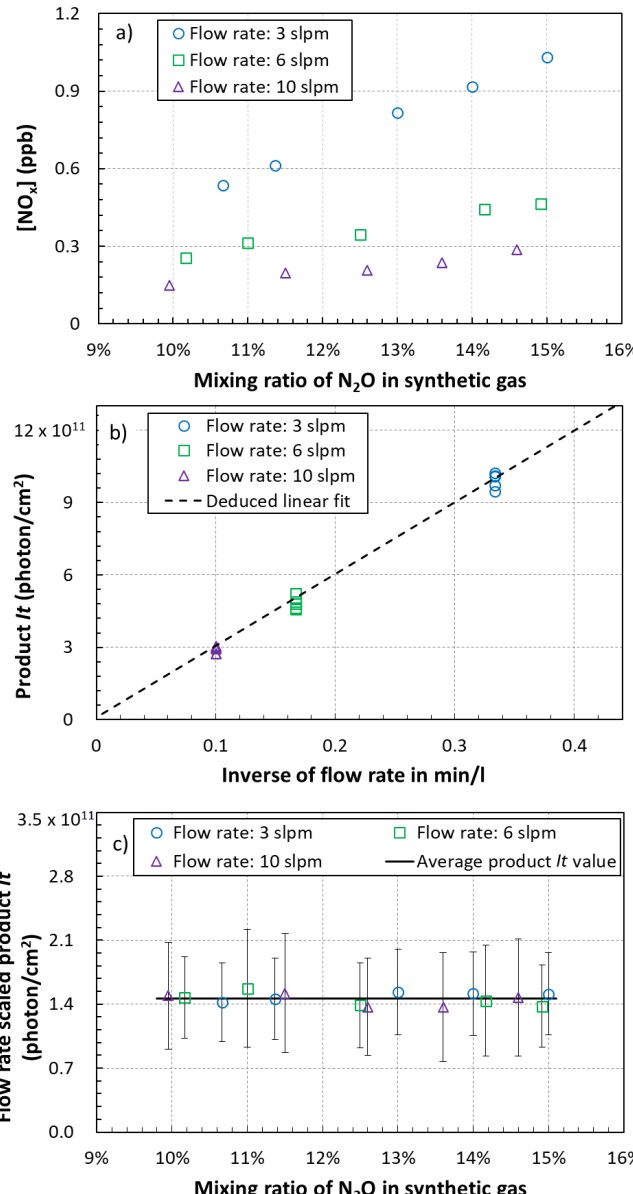

**Figure 3.** The results of $N_2O$ actinometry experiment. a) The produced $NO_x$ concentration as a
function of $N_2O$ mixing ratio. Different colors represent different flow rates. b) The product $It$ as a
function of inverse of flow rate (see detail in text). c) The flow rate scaled product $It$ as a function of
$N_2O$ mixing ratio, which was obtained by scaling product $It$ with the ratio of flow rates (3, 6, and 10
slpm) to 10 slpm. Black line is the average value of flow rate scaled product $It$. The error bars
represent uncertainties ($2\sigma$) for flow rate scaled product $It$. The average uncertainty of the product $It$ is
36%.



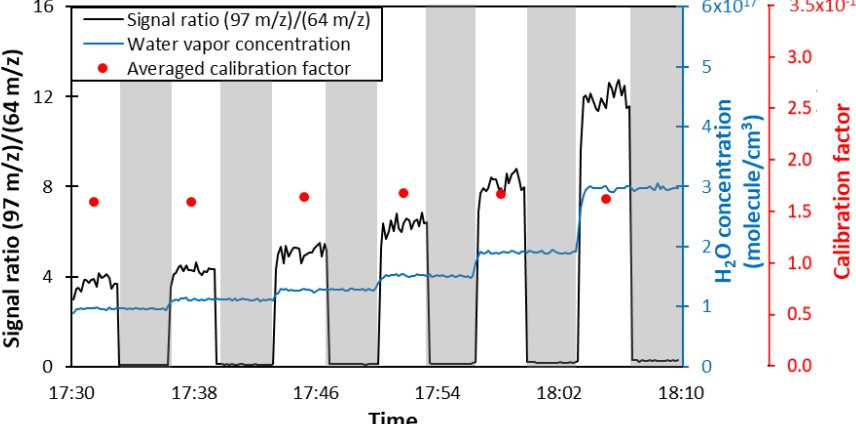

**Figure 4.** The time series showing calibration result. Gray labeled areas represent the background
mode during calibration. Black line represents the ratio of signal intensity at 97 m/z and 64 m/z. Blue
line represents water vapor concentration. Red dots represent the 3-minute averaged calibration
factors at different steps.

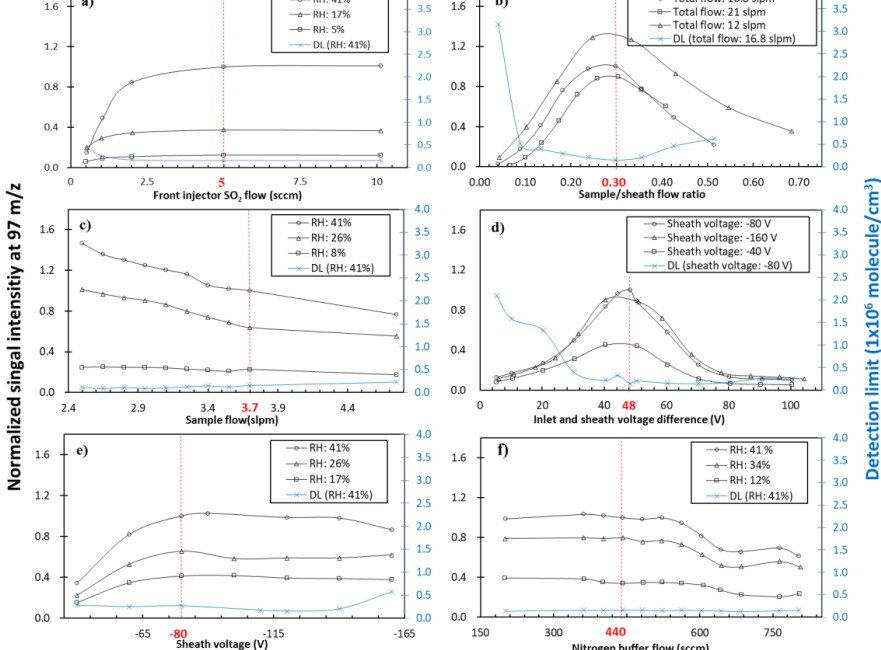



**Figure 5.** The normalized signal intensity at 97 m/z and detection limit as a function of a) $SO_2$ flow
rate b) sample/sheath flow ratio, c) sample flow with fixed sample/sheath flow ratio, d) inlet and
sheath voltages difference, e) sheath voltage with fixed voltage difference between inlet and sheath
voltages, f) $N_2$ buffer flow with the other parameters constant. The signal is normalized based on the
signal intensity at the settings of 5 sccm $SO_2$, 16.8 slpm total flow, 12.6 slpm sheath flow, 3.7 slpm
sample flow, 440 sccm $N_2$ buffer flow, -80V sheath voltage, -32 V inlet voltage, and 41% relative
humidity of the sample air. Red dashed lines highlight the optimized values selected for our CIMS.

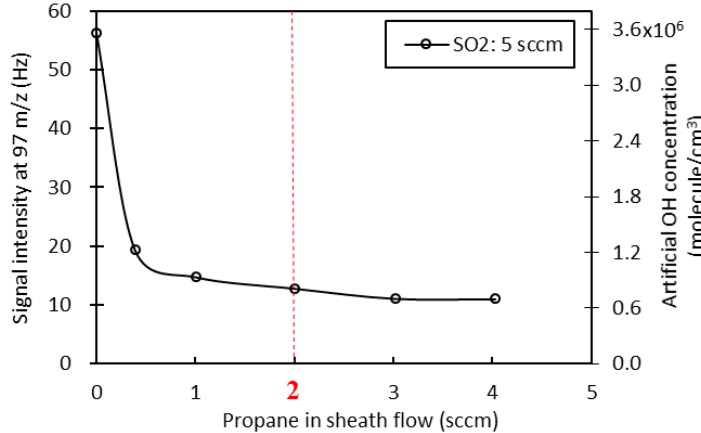

**Figure 6.** Artificial OH signal as a function of propane flow rate added in sheath flow. N2 gas was
used as sample air so that there were no OH radicals in sample air. Red dashed line was the optimized
flow rate applied for our CIMS.

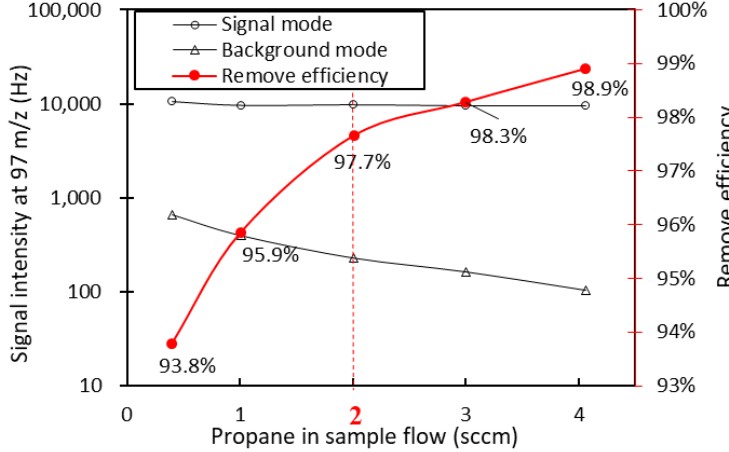

**Figure 7.** Signal intensity at 97 m/z and OH remove efficiency as a function of propane flow rate
added in the sample flow propane during the calibration experiment. The relative humidity of sample
air was 41%. $SO_2$ flow rate was 5 sccm. Red dashed line highlights the optimized flow rate applied for
our CIMS.

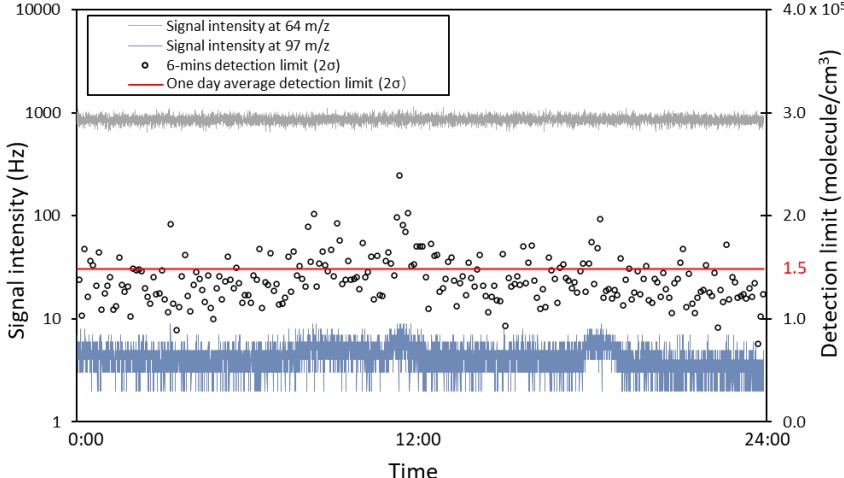

**Figure 8.** The detection limit ($2\sigma$) of the CIMS. One day averaged detection limit is $1.5 \times 10^5$
molecule/$cm^3$.

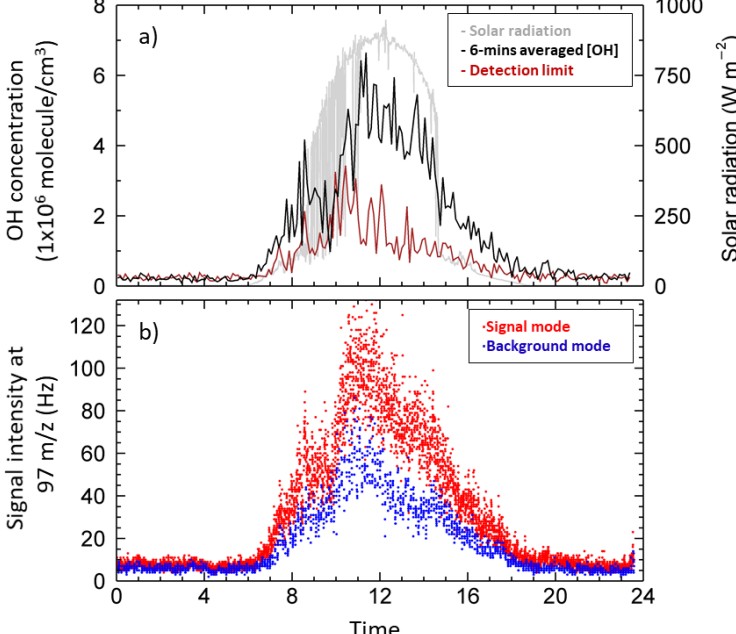

**Figure 9.** a) Diurnal variation of OH concentration and solar radiation on the 11th floor of a teaching
building on the campus of the Hong Kong Polytechnic University on April 25, 2019. b) The signal
intensity at 97 m/z for two different measurement modes.