# Peer review of "Development of a chemical ionization mass spectrometry system for measurement of atmospheric OH radical"

_Atmospheric Measurement Techniques, 2020_

## Referee Comment (RC1) · Anonymous Referee #1 · 3 Aug 2020

Overall evaluation

In this paper, a new Chemical Ionization Mass Spectrometer (CIMS) is described for the measurement of the important atmospheric radical, OH. The results described here are both novel and relevant to the atmospheric chemistry community. The authors describe how difficult this radical is to measure given its short atmospheric lifetime and thus, low concentration. However, due to its critical role in controlling the oxidizing capacity of the atmosphere, accurate quantification of this radial is needed in order to successful evaluate our understanding of atmospheric chemistry. In situ field measurements of OH is relevant in its own right, but accurate assessments of OH these data are also

very valuable in testing atmospheric trace gas models. In this work, the authors present a new CIMS system that is a welcome addition to the relatively few mass spectroscopy-based instrumentation that has been used in the past to measure OH. The present the instrument as an alternative to the more common LIF systems that have been shown to have some drawbacks in accuracy and interference effects in certain environments that the authors describe.

The paper is broken into two sections. The first is a rigorous evaluation of the performance, calibration, sensitivity and optimization of the new instrument in the lab and then second, the work describes a brief field deployment as a "proof of the instrumental concept" that has shown to provide data on the urban atmosphere in a Asian city. The authors report concentrations of OH typically found by other authors in similar environments and describe accuracy and uncertainties in these measurements that are again, typical of those found in the literature. Overall, I found the paper to had excellent technical description of the new CIMS instrument for the measurement of atmospheric OH concentrations and well worth of inclusion in AMT, pending a few small changes and or corrections/clarifications.

Specific Comments or suggested corrections are listed as follows.

Page 3, Line 11. The statement "third most important greenhouse gas" should include a reference.

Page 3, Line 18. The word "enormous" seems unnecessary hyperbole, as does the word "huge" on Page 3, line 21.

Page 8, Line 21. The authors describe sample air entering the system by way of a "blower". I was confused as to the design of this blower. To me, a blower is a plumbing system that makes use of the fan which transfers the air in the emitted environment at high flow or low pressure. I assume the authors are referring to such a system and that is flow rate is regulated by the MFC they mention. I think a clarification of this inlet system would be warranted to describe the initial portion of the inlet more clearly.
Page 9, Line 5. The authors describe the distance of the injectors from "the tube". Is this the front of the inlet scoop with the blower or the front of the ionization section of the instrument?

Page 18, Line 7. The sentence that "Coronal discharges are known to produce OH ions should include references.

Page 19, Line 15. The authors describe the suppression of the instrument sensitivity by the addition of C3F6. Do the authors have any ideas as to why this was so? For the clarity of the paper and for attempts to replicate the results it would be helpful to expand this section with some suggested explanations of the effect of CÂň3F6. Running a simple model of the ion chemistry could shed light on why there is such a suppression of the ion detection system.

Page 19, Section beginning at line 17. Although the addition of NO2 to the system that the authors describe did effect both OH and HO2 the absence of a higher concentration standard that those available to the authors make this section somewhat redundant. I feel removal of this discussion would not detract from the paper at all.

Page 25, line 12. The authors describe the high background signal due to large concentrations of ambient H2SO4. Do they have any feel for what these ambient concentrations were (even ballpark)? Several studies suggest concentrations at sites typical to those the authors measured (e.g. Guo et al., 2012, Zheng et al. 2018) that could be described or at least referenced.

Page 26. The authors describe their future plans to reduce the interference of ambient H2SO4 by using isotopically labeled SO2. This technique was first used by Eisele and Tanner (1991) and this line should thus reference this important earlier paper.

---

## Referee Comment (RC2) · Anonymous Referee #2 · 10 Aug 2020

This paper describes the development of a chemical ionization mass spectrometer for the measurement of OH. The authors describe tests using 210Po, and a corona discharge as ionization sources. They also describe tests using propane, C3F6, and NO2 as different OH scavengers. The authors conclude that using the radioactive 210Po and propane are the best ionization source and OH scavenger respectively. The authors also present a calibration system based upon water photolysis and N2O actinometry.

Overall, I find that there is little novel information presented and the work is not suitable for publication here. The CIMS technique described has been in use since the

early 1990's and the description is a rehash of the works published by Eisele, Tanner, Mauldin, Berresheim, Sjostedt, and others. The calibration is merely an adaptation of that used by Kürtén et al. (2012). The stated limit of detection and uncertainty is approximately the same as reported in previous works.

I find that the present work also lacks detail when compared to the previous works. For example, there are no concentrations of reagent gases given, only flows. Many of the tests performed (varying lens voltages, SO2 and scavenger flows etc.) are specific to the geometry of a particular CIMS system and are typical tests to characterize the instrument and establish operational parameters. The tests of the different ionization sources and OH scavengers are interesting but have little significance when compared to the rest of this work.

On the other hand, a manuscript with some brief descriptions of the measurement techniques and in-depth data analysis of the field measurements would be much stronger and would be of great interest, but will be more suitable for a more general journal (not an instrumentation journal).

Specific questions/comments:

There are many grammatical errors in the manuscript that make it not suitable for publication.

The words titrate and titration are used throughout the manuscript. Titrate and titration are chemistry terms that indicate the smallest amount of reagent necessary to reach an endpoint. The term convert and conversion are more accurate.

No mention is made of the distribution of nitrate ions and clusters. Is this distribution measured? It is shown in Figure 1 that HNO3 is added to the rear injector flow (as in Sjostedt et al. and references therein) in addition to that added to the sheath gas to maintain the HNO3 cluster distribution in the instrument. There is no mention of this aspect in the text. Does the cluster distribution change over time or even between

operating modes (OH signal and background)? Does that affect the sensitivity?

It is stated that the OH background measurement is made to account for pre-existing ambient H2SO4. Is this the only source of the background signal? A publication in Nature by Mauldin et al. states that a significant fraction of this OH background signal caused by a non-OH oxidant of SO2 in environments with large biogenic emissions. Could such emissions affect the sensitivity?

---

## Author Comment (AC1) · 16 Oct 2020

Dear Editor and Referees, Here is the plain text for our response. Please find the original document in the supplement section with color and Figures.

Response to referees' comments

We appreciate all the comments and suggestions by the two referees which have enabled us to improve our manuscript. Please find our itemized responses below and corrections in the re-submitted files. The original comments are italic, and our responses are labeled by blue color, and the revised contents are shown in red color.

[Figure]

The revised content in the manuscript will be highlighted.

Anonymous Referee #1 Overall evaluation In this paper, a new Chemical Ionization Mass Spectrometer (CIMS) is described for the measurement of the important atmospheric radical, OH. The results described here are both novel and relevant to the atmospheric chemistry community. The authors describe how difficult this radical is to measure given its short atmospheric lifetime and thus, low concentration. However, due to its critical role in controlling the oxidizing capacity of the atmosphere, accurate quantification of this radial is needed in order to successful evaluate our understanding of atmospheric chemistry. In situ field measurements of OH is relevant in its own right, but accurate assessments of OH these data are also very valuable in testing atmospheric trace gas models. In this work, the authors present a new CIMS system that is a welcome addition to the relatively few mass spectroscopy based instrumentation that has been used in the past to measure OH. The present the instrument as an alternative to the more common LIF systems that have been shown to have some drawbacks in accuracy and interference effects in certain environments that the authors describe. The paper is broken into two sections. The first is a rigorous evaluation of the performance, calibration, sensitivity and optimization of the new instrument in the lab and then second, the work describes a brief field deployment as a "proof of the instrumental concept" that has shown to provide data on the urban atmosphere in a Asian city. The authors report concentrations of OH typically found by other authors in similar environments and describe accuracy and uncertainties in these measurements that are again, typical of those found in the literature. Overall, I found the paper to had excellent technical description of the new CIMS instrument for the measurement of atmospheric OH concentrations and well worth of inclusion in AMT, pending a few small changes and or corrections/clarifications. Response: Thank you for the positive comments and encouragement.

Specific Comments or suggested corrections are listed as follows. Page 3, Line 11. The statement "third most important greenhouse gas" should include reference. Re-
sponse: An IPCC report is added as a reference for this statement. Reference: Stocker, T. F., Qin, D., Plattner, G., Tignor, M., Allen, S. K., Boschung, J., Nauels, A., Xia, Y., Bex, V., and Midgley, P. (Eds.): IPCC 2013: summary for policymakers in climate change 2013: the physical science basis, contribution of working group I to the fifth assessment report of the intergovernmental panel on climate change, Cambridge University Press, Cambridge, New York, York, USA, 2013.

Page 3, Line 18. The word "enormous" seems unnecessary hyperbole, as does the word "huge" on Page 3, line 21. Response: We have changed the "enormous" and "huge" to "very" and "big", respectively.

Page 8, Line 21. The authors describe sample air entering the system by way of a "blower". I was confused as to the design of this blower. To me, a blower is a plumbing system that makes use of the fan which transfers the air in the emitted environment at high flow or low pressure. I assume the authors are referring to such a system and that is flow rate is regulated by the MFC they mention. I think a clarification of this inlet system would be warranted to describe the initial portion of the inlet more clearly. Response: Thank you for the suggestion. The blower we use here is exactly a plumbing system that transfers the air from the ambient to the front of the sample inlet. Most air in the tube is released back to the ambient. Only a small amount of the air flow from the center of tube flow is drawn into the CIMS by the sample inlet for further measurement (This small amount of flow in the sample inlet is controlled by MFC). Therefore, only high flow speed is required to reduce wall loss in the tube and the MFC is not needed for the blower. For clarification of the function of the blower, the description of the inlet system is modified as follows. Figure 1 also has some changes for the path of the sample air. Please note that the color of this figure is different from the one in the revised version in the manuscript to highlight the blower flow. Revised contents (Page 7):

"As shown in Figure 1, during OH measurements, air sample at ambient temperature and pressure is first drawn into a 5 cm diameter, 32 cm long stainless-steel tube. A

turbulence-reducing scoop is attached to the front of the tube. The flow velocity at the center of the tube is 5 m/s, which is measured manually using a pitot. The central part of the air is then drawn through a 1.6 cm diameter stainless steel inlet into the chemical conversion region with the flow rate being determined by a mass flow controller (MKS, MFC company). The excess flow in the tube is vented back into the atmosphere via the inlet blower."

Page 9, Line 5. The authors describe the distance of the injectors from "the tube". Is this the front of the inlet scoop with the blower or the front of the ionization section of the instrument? Response: Thanks for pointing out the inaccuracy. It should be the inlet for the conversion region not "the tube" with the blower. We have changed "the tube" to "the stainless sample inlet" for a better understanding on page 7.

Page 18, Line 7. The sentence that "Coronal discharges are known to produce OH ions should include references.

Response: Thanks for the suggestion. A previous paper (Kukui et al. 2008) is added. Reference: Kukui, A., Ancellet, G. and Le Bras, G.: Chemical ionization mass spectrometer for measurements of OH and Peroxy radical concentrations in moderately polluted atmospheres, J. Atmos. Chem., 61(2), 133–154, doi:10.1007/s10874-009-9130-9, 2008.

Page 19, Line 15. The authors describe the suppression of the instrument sensitivity by the addition of C3F6. Do the authors have any ideas as to why this was so? For the clarity of the paper and for attempts to replicate the results it would be helpful to expand this section with some suggested explanations of the effect of C3F6. Running a simple model of the ion chemistry could shed light on why there is such a suppression of the ion detection system. Response: Thanks for the suggestion. We re-examined the suppression problem and found that it was due to concurrent measurements of mass at 62 m/z and 64 m/z, the former of which saturated the aged detector. As shown in Figure S2 below, when the detector did not measure the 62 m/z mass, the addition of

C3F6 only caused the 6 m/z signal to drop for a short time. In the revised manuscript, this new figure is used, and the corresponding discussion has been revised.

Revised contents:

"Figure S2. Signal intensity at a) 62 m/z and b) 64 m/z for reagent ion detection by an aged detector when using C3F6 as OH scavenger. a) After the C3F6 suppression, the signal continuously decreases. b) the suppression recovers after time. It is noted that the scale in the y-axis of a) and b) are different." In section 5.2 (Page 15-14): "For C3F6, although its OH removal efficiency was also high, it suppressed the signal intensities detected by the mass detector. This suppression recovered after a time when the detector was new (figure not shown). However, when the detector was aged, such suppression triggered a continuous decrease of the reagent signal as shown in Figure S2a. In this case, the change of reagent ions detection signal can avoid such a decrease (Figure S2b)."

Page 19, Section beginning at line 17. Although the addition of NO2 to the system that the authors describe did effect both OH and HO2 the absence of a higher concentration standard that those available to the authors make this section somewhat redundant. I feel removal of this discussion would not detract from the paper at all.

Response: Thanks for the comments, and we agree. The section of NO2 has been removed.

Page 25, line 12. The authors describe the high background signal due to large concentrations of ambient H2SO4. Do they have any feel for what these ambient concentrations were (even ballpark)? Several studies suggest concentrations at sites typical to those the authors measured (e.g. Guo et al., 2012, Zheng et al. 2018) that could be described or at least referenced. Response: Thanks for the comment and suggestion of the references. The background signal in our instrument is the combination of H2SO4 in the ambient air and the instrumental noise. We calculated the H2SO4 signal by subtracting the instrument noise from the total background, with the instrumental being determined by sampling the synthetic air. The ambient H2SO4 concentrations were estimated in the range of 2-8 x 106 molecule/cm3. We could not find the suggested references because of the very common last names of Guo and Zheng. Revised contents (on Page 19): "The ambient H2SO4 concentration (10-sec average), which was estimated by the normalized background signal divided by the OH calibration factor, varied from 2.6 x 105 in before sunrise to 8 x 106 molecule/cm3 in mid-day (figure not shown)."

Page 26. The authors describe their future plans to reduce the interference of ambient H2SO4 by using isotopically labeled SO2. This technique was first used by Eisele and Tanner (1991) and this line should thus reference this important earlier paper. Response: Thanks for the suggestion. the reference has been added as suggested. Reference: Eisele, F. L. and Tanner, D. J.: Ion-assisted tropospheric OH measurements, edited by Intergovernmental Panel on Climate Change, J. Geophys. Res., 96(D5), 9295–9308, doi:10.1029/91JD00198, 1991.

Anonymous Referee #2 This paper describes the development of a chemical ionization mass spectrometer for the measurement of OH. The authors describe tests using 210Po, and a corona discharge as ionization sources. They also describe tests using propane, C3F6, and NO2 as different OH scavengers. The authors conclude that using the radioactive 210Po and propane are the best ionization source and OH scavenger respectively. The authors also present a calibration system based upon water photolysis and N2O actinometry. Overall, I find that there is little novel information presented and the work is not suitable for publication here. The CIMS technique described has been in use since the early 1990's and the description are a rehash of the works published by Eisele, Tanner, Mauldin, Berresheim, Sjostedt, and others. The calibration is merely an adaptation of that used by Kürtén et al. (2012). The stated limit of detection and uncertainty is approximately the same as reported in previous works. Response: Applying the CIMS technique to measure ambient OH has been a very challenging undertaking. Although its measurement principle was established some time ago, and

several units of the CIMS have been developed in the world, but the technique is by no means mature. The detailed experimental procedures to make a CIMS functional for OH measurement have not been given in previous publications, nor provided with the commercially available CIMS. These have hindered the application of this technique to re-examine the missing OH source issue. To address this problem, we conducted comprehensive tests of a new CIMS, including a comparison of the different ion source and scavenger gas, optimization of sensitivity, and calibration. We believe that our article reporting the detailed experimental results will fill the gap of the missing technical knowledge, and will be useful for the new development/applications of CIMS for measurement of OH and other chemicals. We have shortened descriptions of the basic theory of the OH-CIMS and calibration and emphasize the optimization parts.

I find that the present work also lacks detail when compared to the previous works. For example, there are no concentrations of reagent gases given, only flows. Response: We have added relevant concentrations. The concentration of SO2 and propane in the sample flow is 12 ppm and 535 ppm, respectively. The concentration of the reagent gas (HNO3) is not measured; however, its exact concentration is not required in the measurement. Instead, the ratio of NO3- and HSO4- needs to be measured and used in equation 2 shown on the manuscript to derive OH concentration. We added a short discussion on the necessity of the absolute concentration for reagent gases. Revised content on Page 8: "As discussed in Berresheim et al. (2000), the absolute concentration of the H2SO4Ǎ and reagent ion (NO3-) is not required as the OH concentration is determined based on their relative signal strength and the calibration factor." Reference: Berresheim, H., Elste, T., Plass-Dülmer, C., Eisele, F. L. and Tanner, D. J.: Chemical ionization mass spectrometer for long-term measurements of atmospheric OH and H2SO4, Int. J. Mass Spectrom., 202(1–3), 91–109, doi:10.1016/S1387-3806(00)00233-5, 2000.

Many of the tests performed (varying lens voltages, SO2 and scavenger flows etc.) are specific to the geometry of a particular CIMS system and are typical tests to characterize the instrument and establish operational parameters. Response: We agree with the above comments. However, as we responded to a previous comment, none of the previous papers have detailed descriptions of systematic tests for the optimization of the CIMS. We believe such test procedures and results are helpful for new CIMS developer/operator. Although the set-up parameters are specific to our CIMS unit, the general procedures are applicable to other CIMS. To make our work more applicable to other CIMS, we have added some rationales behind the various tests and rearranged the section. Revised contents in Section 5.4. Instrument sensitivity and noise (Start from Page 15): The sensitivity (S) of the CIMS instrument to the OH radicals depends on the reaction efficiency of OH and SO2 in the chemical conversion region (f(CE)), the conversion efficiency of H2SO4 to ãĂŰ"HSO" ãĂŮ_"4" ˆ"-" in chemical ionization region (f(IE)), and the transmitted efficiency of ãĂŰ"HSO" ãĂŮ_"4" ˆ"-" from sample inlet to the mass spectrometer system (f(TE)): S∼f(CE)Âůf(IE)Âůf(TE)

f(CE) is dependent on the reaction time and the SO2 concentration of the conversion reactions (R1-3). f(IE) is affected by the flow dynamics, which determines the mixing of flows, and the electric field inside the ionization region, which forces the NO_3ˆ- ·(HNO_3 )_m·(HO_2 )_n primary ions to the center of the region for H2SO4 ionization. The f(CE) is related to the N2 buffer and induces an electric field in the pinhole area. In this work, the f(CE) is first optimized for the maximum conversion of the ambient OH to H2SO4 by adjusting the SO2 flow and the sample flow rate. Then, to achieve the best f(IE) for H2SO4 ionization, the flow dynamic and electric field are optimized by adjusting the sample/sheath flow ratio and the voltages applied to sample and sheath flow. Finally, the N2 buffer flow rate and the voltages of the pinhole are adjusted to control the f(TE) to determine the amounts of ions entering the detector. During the optimization, the calibration tube is applied to produce OH radicals and control its concentration. 5.4.2 Conversion efficiency Figure 5a shows the normalized signal intensity (NSI) at 97 m/z for ãĂŰ"HSO" ãĂŮ_"4" ˆ"-" as a function of the flow rate of SO2 (0.9 vol.%). The NSI first increased with increased SO2 and reached a stable level at a flow rate > ∼2.5 sccm, which did not vary with the relative humidity. This result indicates

that the SO2 concentration at the flow rate of 2.5 sccm was adequate to convert sampled OH to H2SO4. Since the concentration of OH radicals produced by the calibration unit during optimization was 1 to 2 magnitude higher than in that in ambient condition, the 2.5 sccm flow of SO2 is adequate for ambient measurement. We set the SO2 flow rate at 5 sccm with a factor of 2 margins, following the previous study (Sjostedt et al., 2007). With this flow rate, the concentration of SO2 in the sample flow is 12 ppm. The effects of the sample flow rate on NSI are shown in Figure 5b. During adjusting the sample flow rate, if the sheath flow rate remains the same, the conversion time and the flow dynamics will be affected. Thus, in order to show the effect of conversion time to NSI only, the sheath flow was adjusted along with the sample flow to maintain the sample/sheath flow ratios and control the f(IE) in Figure 5b. Briefly, the NSI increased with the decrease of sample flow rate, which can be explained by a longer OH conversion time (R1-3) and a higher f(CE) at a lower flow rate. However, the increased reaction time will also increase the OH interference produced from HO2 recycling in the presence of NO in sample air. Previous studies usually kept the reaction time less than 60 ms to mitigate such interference (e.g. Tanner et al., 1997). After considering the reaction time and interference, the sample flow rate was set at 3.7 slpm, which gives a reaction time of ∼47 ms. After the above selection of the SO2 concentration and sample flow rate, the optimal f(CE) is determined. 5.4.3 Ionization efficiency Figure 5c shows the NSI as a function of the ratio of sample flow to sheath flow in the ionization region. The NSI firstly increased and then decreased with the increased ratio, with a peak value at a sample/sheath flow ratio of 0.3. This optimized ratio was independent of the total flow rates from 12 to 21 slpm. This ratio produced a turbulent flow in the chemical ionization region. Such flow dynamics facilitates a fast mixing of the reactants and enhances the f(IE) of $H_2SO_4^-$ as well as the NSI at 97 m/z (Tanner and Eisele, 1995; Tanner et al., 1997). Figure 5d-e shows the effects of voltages applied to the sample and sheath flow on NSI. The NSI first increased and then decreased as the increase of difference in voltage between the sample and sheath flow (figure 5d). At the voltage difference of 48 V, the peak NSI was achieved, and this voltage difference is

selected. Figure 5e shows the NSI increased with the negative sheath voltage and then kept stable with sheath voltage < -70V. This shows that when it is negatively charged, it is the voltage difference but not the exact voltage that affects the NSI. In our studies, we set the inlet and sheath voltages at -32 and -80 V, respectively. The cross interactions of sample/sheath flow and voltages on NSI were also evaluated (see Figure S3). The result shows that the highest NSI was achieved when the sample/sheath flow ratio was close to 0.3, independent of the voltages. The optimized f(IE) of the CIMS is achieved by the above selections of the flow ratio and electric field. 5.4.4 Transmission efficiency The N2 buffer flow rate controls the proportion of sample air in dry N2 (refer to figure 1), thereby affecting the amount of ion clusters in the sample air entering the mass detector. Figure 5f shows that the NSI increased with the decreased buffer flow rate, as expected. However, a lower flow rate of N2 buffer gas also allows more undesired neutral molecules and particles in sample air to enter the mass spectrometer (Berresheim et al., 2000). With these considerations, the flow rate of N2 buffer gas was set as 440 sccm. To force the ions to the center of the pinhole, the voltage applied to before and on the pinhole was set at -70V and -40V, respectively. 5.4.5 Noise minimization After the optimization of CIMS's sensitivity, noise minimization is needed to reduce the signal that is not related to the ambient OH concentration. As discussed above, the noises for OH measurements include H2SO4 in ambient air and artificial OH produced by the ion source. These noises can be mitigated by adding a scavenger gas in the sheath flow to eliminate artificial OH and in the sample flow to quantify ambient H2SO4 (Figure 1). Below we determine the optimal flow rates for the scavenger gas. Figure 6 shows the HSO4- signal intensity as a function of propane flow rates in sheath flow, with N2 gas as the sample air. When propane was not added, the artificial OH concentration from the 210Po ion source was $\sim 3.5 \times 10^6$ molecules cm-3, which is comparable to the typical OH concentrations in ambient environments. When propane was added into the sheath gas, the artificial signals were reduced with the increasing propane flow and kept stable at $\sim 1 \times 10^6$ molecules cm-3 when the flow rate was higher than 1 sccm. We thus set a flow rate of 2 sccm for propane in the sheath flow. Figure 7 shows the

removal efficiency (RE) of OH by propane added in the sample flow as a function of the propane flow rates. The OH radicals were produced by the calibration unit described in Section 4.2. The RE increased with the increased propane flow rate initially and leveled off at the flow rate > 1 sccm. We adopt the flow rate of propane of 2 sccm (~535 ppm), which led to ~98% removal efficiency for OH. As OH concentrations in this test are much higher than those in typical ambient air, the RE of the propane for ambient OH should be even large at the selected flow rate.

The tests of the different ionization sources and OH scavengers are interesting but have little significance when compared to the rest of this work. On the other hand, a manuscript with some brief descriptions of the measurement techniques and in-depth data analysis of the field measurements would be much stronger and would be of great interest, but will be more suitable for a more general journal (not an instrumentation journal).

Response: As responded to the previous comments, the descriptions of comprehensive tests on OH scavengers (and other components) have not been given in previous studies, and we believe that they are helpful to new CIMS users and decide to keep them. The NO2 section has been deleted in response to the first reviewer's comment.

Specific questions/comments: There are many grammatical errors in the manuscript that make it not suitable for publication. Response: Thanks for the comment, the manuscript has been carefully revised to eliminate the grammatical errors.

The words titrate and titration are used throughout the manuscript. Titrate and titration are chemistry terms that indicate the smallest amount of reagent necessary to reach an endpoint. The term convert and conversion are more accurate. Response: Thanks for the comment on specific term use. The words titrate and titration has been widely used in previous CIMS articles for the same purpose. (Muller et al., 2018; Tanner et al., 1997; Berresheim et al., 2003; Kukui et al., 2014; Mauldin et al., 1999; Acker et al., 2006) However, after considering the meaning of "titrate" and "titration" provided by

the reviewer, we agree it is better to use "convert" instead of "titrate". And the "titration efficiency" was also changed to the "conversion efficiency".

No mention is made of the distribution of nitrate ions and clusters. Is this distribution measured? It is shown in Figure 1 that HNO3 is added to the rear injector flow (as in Sjostedt et al. and references therein) in addition to that added to the sheath gas to maintain the HNO3 cluster distribution in the instrument. There is no mention of this aspect in the text. Does the cluster distribution change over time or even between operating modes (OH signal and background)? Does that affect the sensitivity?

Response: Thanks for pointing out the rear HNO3. We have added a description of the rear HNO3. The change of mode will not affect the concentration of HNO3 in the system because the HNO3 is added from the rear injector continuously in both modes, and the pulsed flow of scavenger is compensated by the nitrogen gas at the same flow rate. Thus, the sensitivity will not be affected. Addition Content in Section 3.1.2 Chemical ionization region (Page 8): "Additionally, the N2 carried HNO3 is also added through the rear injector to maintain the ion cluster distribution and further improve the stability of the reagent ion signal (Sjostedt et al., 2007)."

It is stated that the OH background measurement is made to account for pre-existing ambient H2SO4. Is this the only source of the background signal? A publication in Nature by Mauldin et al. states that a significant fraction of this OH background signal caused by a non-OH oxidant of SO2 in environments with large biogenic emissions. Could such emissions affect the sensitivity?

Response: The suggested Nature article described a possible source for H2SO4 from the stabilized Criegee intermediate or its derivatives which are linked to the presence of alkenes or biogenic origin. However, the field measurement presented by our manuscript was done on the 11th floor (50 m above the ground) next to the highway. In this case, the biogenic emissions are relatively low compare to anthropogenic emission. Therefore, the H2SO4 signal in our manuscript is mostly caused by the

anthropogenic emission.

Please also note the supplement to this comment:
https://amt.copernicus.org/preprints/amt-2020-252/amt-2020-252-AC1-
supplement.pdf

———————————————

[Figure]

[Figure]

**Fig. 1.** Figure 1

**Fig. 2.** Figure S2

---

## Author Comment (AC3) · 17 Oct 2020

[revised manuscript text omitted]

Despite the previous development and application of CIMS for OH measurements, applications of this technique remain challenging, and are hindered by a lack of detailed experimental procedures to make a CIMS functional. And there have been limited deployments of CIMS for ambient OH determination compared to the LIF technique, and there has been no report of OH measurements by the CIMS method in Asia. In the present work, we describe here a new CIMS system that has been tested and optimized at The Hong Kong Polytechnic University (PolyU). The instrument was built at THS, Inc (Atlanta, Georgia) with the same design as the CIMS from the group of Eisele and Tanner. The measurement principles, configurations of the CIMS instrument, and a calibration unit are described in detail. Different scavenger gases, ion sources, and primary ions detection was compared. In addition, the sensitivity and noise of the CIMS instrument to OH radicals were tested by optimizing the flow rates and voltages. Accordingly, their optimal settings were derived. Finally, the initial measurement of ambient OH measurement was presented. The results presented in this work provide detailed technical information for other researchers who wish to apply the CIMS to ambient OH measurement. To our knowledge, this instrument is the first OH measuring CIMS in Asia.

**2. Measurement principles**

The measurement of hydroxyl radical (OH) in this study was made with a chemical ionization mass spectrometry (CIMS) technique, which has been described previously (Tanner et al., 1997; Sjostedt et al., 2007). Briefly, the ambient OH is converted to $H_2SO_4$ by adding $SO_2$ into the sample air flow, which initiates the following reaction sequence in the presence of oxygen and water vapor:

$$OH + SO_2 + M \rightarrow HSO_3 + M \qquad\qquad (R1)$$

$$HSO_3 + O_2 \rightarrow SO_3 + HO_2 \qquad \text{(R2)}$$

$$SO_3 + 2H_2O \rightarrow H_2SO_4 + H_2O \qquad \text{(R3)}$$

To mitigate interference by ambient $H_2SO_4$, a scavenger gas is periodically added into the sample air flow to remove OH radicals. Then, $H_2SO_4$ produced from the reaction of OH and

$SO_2$ can be obtained:

$$[H_2SO_4]_{OH} = [H_2SO_4]_{TS} - [H_2SO_4]_{BS} \qquad \text{(E1)}$$

$[H_2SO_4]_{TS}$ and $[H_2SO_4]_{BS}$ are $H_2SO_4$ concentrations with and without adding scavenger gas in the front injector, respectively. (See Section 3.1.1 for details)

Apart from the interference from the pre-existing $H_2SO_4$, the reactions of NO with ambient or

Rection 2 produced peroxy radicals ($HO_2 + RO_2$) can produce OH in the sample flow (Sjostedt et al., 2007):

$$RO_2 + NO + O_2 \rightarrow R^{'}CHO + HO_2 + NO_2 \qquad \text{(R4)}$$

$$HO_2 + NO \rightarrow OH + NO_2 \qquad \text{(R5)}$$

To reduce the positive bias from Reaction 5, another scavenger gas is added into the sample flow after $SO_2$ to scavenge recycled OH radicals.

The $H_2SO_4$ is then converted into $HSO_4^-$ by chemical ionization in reaction with the $NO_3^-$

primary reactant ions:

$$H_2SO_4 + NO_3^- \cdot (HNO_3)_m \cdot (H_2O)_n \rightarrow HSO_4^- \cdot (HNO_3)_m (H_2O)_n + HNO_3 \qquad \text{(R6)}$$

$NO_3^- \cdot (HNO_3)_m \cdot (H_2O)_n$ are cluster ions with m and n mostly of 0-2 and 0-3 (Berresheim et al., 2000). These cluster ions are generated by the reaction of $HNO_3$ vapor with electrons (Fehsenfeld et al., 1975):

$$HNO_3 + e^- \rightarrow NO_2^- + OH \qquad \text{(R7)}$$

$$HNO_3 + NO_2^- \rightarrow NO_3^- + HONO \qquad \text{(R8)}$$

$$NO_3^- + (HNO_3)_m + (H_2O)_n + M \rightarrow NO_3^- \cdot (HNO_3)_m \cdot (H_2O)_n + M \qquad \text{(R9)}$$

Where $e^-$ is emitted from an ion source. The OH radical (artificial OH) formed from primary ion creation (Reaction 7) is not desirable and regards as noise signal, see details in Section

5.4.3. The  ion clusters are subsequently dissociated by the collisional dissociation chamber (CDC):

$$NO_3^- \cdot (HNO_3)_m \cdot (H_2O)_n + M \rightarrow NO_3^- + (HNO_3)_m + (H_2O)_n + M \qquad \text{(R10)}$$

$$HSO_4^- \cdot (HNO_3)_m \cdot (H_2O)_n + M \rightarrow HSO_4^- + (HNO_3)_m + (H_2O)_n + M \qquad \text{(R11)}$$

The OH is finally detected by a mass spectrometer system as $HSO_4^-$ at 97 m/z.

**3. CIMS system**

Figure 1 shows the schematic of our CIMS system, which including a sample inlet system and a mass spectrometer system. The sample inlet system has two regions: chemical conversion region and chemical ionization region. The chemical conversion region is where $H_2SO_4$ formed by the conversion reaction of OH and $SO_2$. Then the $H_2SO_4$ converted in to $HSO_4^-$ ion cluster in chemical ionization region. The mass spectrometer system consists of three parts including a collisional dissociation chamber (CDC) for ion cluster dissociation, an ion guide chamber (IGC) to refocused the ions, and an ion detection chamber (IDC).

**3.1. Sample inlet**

As shown in Figure 1, during OH measurements, air sample at ambient temperature and pressure is first drawn into a 5 cm diameter, 32 cm long stainless-steel tube. A turbulence-reducing scoop is attached in the front of the tube. The flow velocity at center of the tube is 5 m/s, which is measured manually using a pitot.  The central part of the air is then drawn through a 1.6 cm diameter stainless steel inlet into the chemical conversion region with the flow rate being determined by a mass flow controller (MKS, MFC company). The excess flow in the tube is vented back into the atmosphere via the inlet blower.

**3.1.1 Chemical conversion region**

The chemical conversion region in Figure 1 is equipped with two pairs of stainless steel needle injectors that are placed in opposite positions. The first (front injectors) pair is installed at a 69 mm distance from the stainless sample inlet. The distances between the first (front) and second (rear) pairs are 25.8 mm. To measure OH radicals, $SO_2$ is continuously added into the sample flow at the front injectors to convert OH into $H_2SO_4$ (Reactions 1-3). The purity of $SO_2$ is 0.9 vol.%.

As discussed above, to eliminate the ambient $H_2SO_4$ interference, another flow is added through a zero-dead space four-way electrically operated valve, which is automatically switched the injection positions of scavenger gas and pure $N_2$ every 3 minutes (see the pulsed flow in Figure 1). When the scavenger gas is added through the front injectors to the sample flow, $N_2$ is switched through the rear injectors. The CIMS is then running in background mode. Under this condition, atmospheric OH simultaneously reacts with $SO_2$ and the scavenger gas, with the reaction of OH with scavenger gas being much faster than $SO_2$. This configuration produces background signal (BS) from the interferences of atmospheric $H_2SO_4$ and the ion source, with negligible contribution from atmospheric OH. When the scavenger gas and $N_2$ are switched into the sample flow through the rear and front injectors, respectively, CIMS is running in the signal mode. Atmospheric OH is all converted by $SO_2$ and the total signal (TS) is produced. In addition, another flow of scavenger gas is added continuously into the sample flow through the rear injectors to scavenge OH radicals generated from Reaction 5. The OH concentration is obtained from the ratio of the difference between the total signal and the background signal to the primary ion ($NO_3^-$) signal. (Tanner and Eisele, 1995):

$$[OH] = \frac{1}{C} \times \frac{\{HSO_4^-\}_{TS} - \{HSO_4^-\}_{BS}}{\{NO_3^-\}} \tag{E2}$$

Where the square brackets and text braces denote concentrations and signal counts, respectively. $C$ is the calibration factor. As discussed in Berresheim et al. (2000), the absolute concentration of the $H_2SO_4$ and reagent ion ($NO_3^-$) is not required as the OH concentration is determined based on their relative signal strength and the calibration factor.

**3.1.2 Chemical ionization region**

The sample flow through the chemical conversion region is then drawn into the chemical ionization region and mixed with the sheath gas (Figure 1). The sheath flow is continuously drawn into the same region through an annular space between 3.5 cm o.d. and 1.2 cm o.d. stainless steel tubes by a diaphragm pump (KNF-813). These tubes are concentric with the downstream end of the chemical conversion region. The sheath gas is produced by a zero-air generator (Thermo Electron Corporation, Model 111) attached with active charcoal and silica gel to remove trace gases such as $SO_2$ and $NO_x$. Before entering the ionization region, $HNO_3$ vapor and the scavenger gas are added continuously to the sheath gas. The $HNO_3$ vapor is obtained by $N_2$ carrier gas passing through the headspace of a reservoir of concentrated liquid $HNO_3$. When $HNO_3$ doped sheath gas passes through the ion source (Figure 1), $NO_3^- \cdot$ $(HNO_3)_m \cdot (H_2O)_n$ reactant ions are produced by the reaction of $HNO_3$ and electrons (Reactions 7-9). Additionally, the $N_2$ carried $HNO_3$ is also added through the rear injector to maintain the ion cluster distribution and further improve the stability of the reagent ion signal (Sjostedt et al., 2007).

The $NO_3^- \cdot (HNO_3)_m \cdot (H_2O)_n$ reactant ions then react with $H_2SO_4$ molecules from the sample air to form $HSO_4^- \cdot (HNO_3)_m (H_2O)_n$ cluster ions in the chemical ionization region according to Reaction 6. Voltages are applied to the sample and sheath flow tubes to produce an electrical field to force the reactant ions to the center of the chemical ionization region and enhance the interaction of reactant ions with $H_2SO_4$. The optimization of voltages for better sensitivity is shown in Section 5.4.2.3.

[revised manuscript text omitted]

**4.3.2 $It$ quantification**

The product $It$ is determined based on the chemical actinometry method (Figure 2b). This method measures NO$_x$ generated from $N_2O$ photolysis with the same calibration unit under the same condition as that for the CIMS calibration. Since $N_2O$ photolysis and $H_2O$ photolysis require the same photon intensity (184.9 nm), product $It$ of $H_2O$ photolysis can be determined by measured NO$_x$ and $N_2O$ mixing ratios produced by $N_2O$ photolysis (Edwards et al., 2003).

Briefly, high purity $N_2O$ (99.9%) mixed with dry synthetic gas flows into the calibration unit.

The photolysis of $N_2O$ generates NO$_x$ through the following reactions (Edwards et al., 2003):

$$N_2O + h\nu \ (184.9 \ nm) \rightarrow N_2 + O(^1D) \tag{R14}$$

$$O(^1D) + O_2 \rightarrow O(^3P) + O_2 \tag{R15}$$

$$O(^1D) + N_2 \rightarrow O(^3P) + N_2 \tag{R16}$$

$$O(^3P) + O_2 + M \rightarrow O_3 + M \tag{R17}$$

$$O(^1D) + N_2O \rightarrow 2NO \tag{R18}$$

$$O(^1D) + N_2O \rightarrow N_2 + O_2 \tag{R19}$$

The $O_3$ produced from Reaction 17 could oxidize NO to NO$_2$. Therefore, the photolysis of $N_2O$

eventually converts it to NO$_x$ which is concurrently measured by a commercial NO$_x$ detector (Thermo, Model 42i-TL) The combined product $It$ is a function of the mixing ratios of $N_2O$,

$N_2$, $O_2$, and produced $NO_x$:

$$It = \frac{(K_{15} \times [O_2] + K_{16} \times [N_2] + (K_{18} + K_{19}) \times [N_2O]) \times [NO_X]}{2 \times K_{18} \times \sigma_{N_2O} \times \phi_{N_2O} \times [N_2O]^2} \quad \text{(E4)}$$

Where $K_{15}$, $K_{16}$, $K_{18}$, $K_{19}$ are the rate constants of Reaction 15, 16, 18 and 19, respectively.

$\sigma_{N_2O}$ is the absorption cross-section of $N_2O$ and $\phi_{N_2O}$ is the photolysis quantum yield. The values for them can be found in previous study (Kurten et al., 2012).

Ideally, the $N_2O$ actinometry experiment should be conducted with the same flow rate as in the

$H_2O$ photolysis experiment such that the reaction time can be the same. However, at the flow rate suitable for CIMS calibration (10 slpm), the concentration of $NO_x$ produced from $N_2O$

photolysis is near the detection limit of the $NO_x$ detector. Hence, the $N_2O$ actinometry experiment was carried out at a lower flow rate (3 and 6 slpm) to increase the reaction time for photolysis and then the $NO_x$ production. The $It$ values for lower flow rate ($It_{HLOW}$) and higher flow rate ($It_{HIGH}$) have the following relationship:

$$It_{HIGH} = \frac{\text{FR}_{LOW} \times It_{LOW}}{\text{FR}_{HIGH}} \quad \text{(E5)}$$

where $\text{FR}_{LOW}$ and $\text{FR}_{HIGH}$ represent different flow rate. Based on this equation, $It_{HIHG}$ can be obtained by scaling $It_{LOW}$ with the ratio of $\text{FR}_{LOW}$ and $\text{FR}_{HIGH}$. The E5 is validated in the next section.

**4.3.2 *It* determination**

Figure 3 shows the results of $N_2O$ actinometry experiment. Figure 3a shows the $NO_x$ produced as the function of $N_2O$ mixing ratios from 10% to 15% at different flow rates ($\text{FR}_{N_2O} = 3, 6$, and 10 slpm). Generally, an increase in the $N_2O$ mixing ratio or a decrease in reaction time (lower flow rate) led to more production of $NO_x$. In figure 3b, the product $It$ corresponding to different flow rate was calculated according to E4 based on the result in Figure 3a. The product

$It$ linearly increased with the inverse of the flow rate, which validates the linear dependency between product $It$ and inverse of the flow rate shown by E5. This linear dependency is consistent with the result in Kurten et al. (2012). In addition, the product $It$ was independent of the $N_2O$ mixing ratios in range of 10% to 15% (Figure 3c). Based on the E5, the flow rate scaled $It$ ($It_{HIGH}$) is calculated from $It_{LOW}$ in Figure 3b multiplying the ratio of $\text{FR}_{LOW}$ (3, 6, and 10 slpm, respectively) to $\text{FR}_{HIGH}$ (10 slpm). The $It$ varied from 1.37 to 1.53 x $10^{11}$ at different flow rates and $N_2O$ mixing ratios. The mean value of $1.46 \times 10^{11}$ photon cm$^{-1}$ was adopted for $It_{HIGH}$.

**4.3.3 Calibration result**

Figure 4 shows an example of a typical procedure for determining the calibration factor. The instrument signals were continuously measured by adjusting $H_2O$ concentrations without changing other parameters. The different OH concentrations were calculated according to E3. For each step, the signal intensities (in Hz) of $HSO_4^-$ and $NO_3^-$ were collected for 6 minutes with 3 minutes each for background mode and signal mode. The calibration factors were determined from the calculated OH concentrations and signal intensities based on E2. The red dots in Figure 4 represent the average calibration factors for every 6 minutes. The result shows that the calibration factors  different steps were very close, ranging from 1.60 to $1.69 \times 10^{-10}$, and were independent of water vapor concentrations. The averaged calibration factor for our CIMS is $1.64 \times 10^{-10}$ molecule/cm$^{-3}$.

**5. CIMS optimizations**

As shown in Figure 1, the CIMS system is complicated, and its performance is affected by different parameters and components. In this section, we present the results of tests for different types of ion sources and scavenger gases (propane and $C_3F_6$), comparison of primary ions detection, and optimization of the instrument sensitivity and noise.

**5.1. Ion source**

Radioactive ion source ([210]Po or [241]Am) and corona discharge source (corona ionizer) have been used as the ion source in previous studies (Berresheim et al., 2000; Sjostedt et al., 2007; Kukui et al., 2008). In this study, [210]Po and corona sources were compared.

[210]Po emitted alpha particles that interact with the carrier gas to quickly form thermalized electrons and positive ions (Fehsenfeld et al., 1975). The formed electrons react with $O_2$ and then $HNO_3$ to produce $NO_3^- \cdot (HNO_3)_m \cdot (H_2O)_n$ reactant ions. [210]Po was used due to its low OH interference and ease of installation. Corona ionizer generates $NO_3^-$ by discharge formed between a tungsten needle and a 1 mm diameter plate 3 mm from the needle tip (Kukui et al., 2008). The corona source has the advantage of producing much higher concentrations of $NO_3^- \cdot (HNO_3)_m \cdot (H_2O)_n$ primary ions compared with radioactive [210]Po or [241]Am foils, which leads to higher concentrations of $HSO_4^-$ and higher signal intensities (and higher sensitivities). However, the corona discharge source is known to produce a significant amount of neutral species including OH radicals (artificial OH), which means the noise is relatively high (Kukui et al., 2008). We compared a $^{210}$Po ions source and a corona source (Figure S1). The result showed that the detection limit of the CIMS with the $^{210}$Po ion source was lower than that with the corona source due to larger noise in the corona source. We previously applied the corona source in the same CIMS to measure peroxy radicals ($RO_2$ and $HO_2$), and the noise level was acceptable compared to the ambient concentration of peroxy radicals. For OH measurement, although a scavenger gas was added in the sheath flow to remove most artificial OH radicals, the remaining interferences were still high compared to ambient OH concentrations.

In this study, $^{210}$Po foils were chosen as the ion source in our CIMS system. It should be noted, however, that a radioactive source like $^{210}$Po often subjects to strict health safety regulations, and the users need apply a permit to use and transfer the radioactive source. In addition, $^{210}$Po undergoes alpha decay to stable $^{206}$Pb with a half-life of about 140 days. Therefore, in order to keep stable signal intensities for primary ions, the $^{210}$Po foils need to be replaced regularly.

**5.2 Scavenger gases**

As discussed in Section 3, scavenger gas is very important for OH measurement by CIMS. Propane, $C_3F_6$, and NO2 have been used by different groups (Berresheim et al., 2000; Sjostedt et al., 2007; Kukui et al., 2008). However, there have been no reports on comparisons of these three scavenger gases. In this study, The first two were compared. (We could not purchase high purity $NO_2$ due to health safety restriction in Hong Kong). The scavenger gas was added in two positions for different purposes. In the sheath flow, the scavenger gas reduced the interference of artificial OH from the ion source; in the sample flow, it terminated the OH conversion (in the rear injector) and eliminated the ambient OH to determine background (in the front injector).

Figure 6 shows that 99.95 vol.% pure propane (Harvest Wise Gases (H.K.) Limited) added in sheath flow could effectively (~80%) remove artificial OH radicals from the ion source with the remaining contributing a low and stable signal at 97 m/z. For the OH removal efficiency of in the sample air, propane could remove OH at 97.7% of OH even at the concentrations of two orders of magnitude higher than ambient OH level (for more details, see Section 5.4.3). The signal intensity of the primary ions was not affected by the added propane and kept stable (Figure S2).

For $C_3F_6$, although its OH removal efficiency was also high, it suppressed the signal intensities detected by the mass detector. This suppression recovered after time when the detector was new (figure not shown). However, when the detector was aged, such suppression triggered a continuous decrease of the reagent signal as shown in Figure S2a. In this case, the change of reagent ions detection signal can avoid such decrease (Figure S2b).

Based on the above test results and considering that propane is inexpensive and easy to purchase, propane was selected as the scavenger gas for our CIMS.

**5.3 Primary ions detection**

Determination of the OH concentrations needs to use the signal intensities of $NO_3^-$ ions according to E2. Some previous researchers traced the $NO_3^-$ ions on the signal intensities at 62

m/z (Kukui et al., 2012; Sjostedt et al., 2007; Tanner et al., 1997). We found that the concentrations of $NO_3^-$ in the inlet system were extremely high. Even though a small portion of the $NO_3^-$ ions were finally detected by the mass detector, the signals were still very strong. After operating the CIMS with detecting the signal of $NO_3^-$ ions at 62 m/z about half a year, all signals from the channeltron detector dropped significantly which may be due to the accelerated aging of the detector by the high concentrations of $NO_3^-$ ions. Therefore, the isotopic signal ($N^{18}O_3^-$)

at 64 m/z was chosen to detect $NO_3^-$ primary ions for extended operation. The signal intensity at 64 m/z is lower than at 62 m/z by about a factor of 167.

**5.4 Instrument sensitivity and noise**

**5.4.1 Sensitivity optimization**

The sensitivity (S) of the CIMS instrument to the OH radicals depends on the reaction efficiency of OH and $SO_2$ in chemical conversion region (f(CE)), the conversion efficiency of

$H_2SO_4$ to $HSO_4^-$ in chemical ionization region (f(IE)), and the transmitted efficiency of $HSO_4^-$

from sample inlet to mass spectrometer system (f(TE)):

$$S \sim f(CE) \cdot f(IE) \cdot f(TE)$$

f(CE) is dependent on the reaction time and the $SO_2$ concentration of the conversion reactions (R1-3). f(IE) is affected by the flow dynamics, which determines the mixing of flows, and the electric field inside the ionization region, which forces the $NO_3^- \cdot (HNO_3)_m \cdot (HO_2)_n$ primary ions to the center of the region for $H_2SO_4$ ionization. The f(CE) is related to the $N_2$ buffer and the induces electric field in the pinhole area. In this work, the f(CE) is first optimized for the maximum conversion of the ambient OH to $H_2SO_4$ by adjusting the $SO_2$ flow and the sample flow rate. Then, to achieve best f(IE) for $H_2SO_4$ ionization, the flow dynamic and electric field are optimized by adjusting the sample/sheath flow ratio and the voltages applied to sample and sheath flow. Finally, the $N_2$ buffer flow rate and the voltages of the pinhole are adjusted to control the f(TE) to determine the amounts of ions entering the detector. During the optimization, the calibration tube is applied to produce OH radical and control its concentration.

**5.4.2 Conversion efficiency**

Figure 5a shows the normalized signal intensity (NSI) at 97 m/z for $HSO_4^-$ as a function of the flow rate of $SO_2$ (0.9 vol.%). The NSI first increased with increased $SO_2$, and reached a stable level at a flow rate > ~2.5 sccm, which did not vary with the relative humidity. This result indicates that the $SO_2$ concentration at the flow rate of 2.5 sccm was adequate to convert sampled OH to $H_2SO_4$. Since the concentration of OH radical produced by the calibration unit during optimization was 1 to 2 magnitude higher than in that in ambient condition, the 2.5 sccm flow of $SO_2$ is adequate for ambient measurement. We set the $SO_2$ flow rate at 5 sccm with a factor of 2 margin, following the previous study (Sjostedt et al., 2007). With this flow rate, the concentration of $SO_2$ in sample flow is 12 ppm.

The effects of the sample flow rate on NSI are shown in Figure 5b. During adjusting the sample flow rate, if the sheath flow rate remains the same, the conversion time and the flow dynamics will be affected. Thus, in order to show the effect of conversion time to NSI only, the sheath flow was adjusted along with the sample flow to maintain the sample/sheath flow ratios and control the f(IE) in Figure 5b. Briefly, the NSI increased with the decrease of sample flow rate, which can be explained by a longer OH conversion time (R1-3) and a higher f(CE) at a lower flow rate. However, the increased reaction time will also increase the OH interference produced from $HO_2$ recycling in the presence of NO in sample air. Previous studies usually kept the reaction time less than 60 ms to mitigate such interference (e.g. Tanner et al., 1997). After considering the reaction time and interference, the sample flow rate was set at 3.7 slpm, which gives a reaction time of ~47 ms. After the above selection of the $SO_2$ concentration and sample flow rate, the optimal f(CE) is determined.

**5.4.3 Ionization efficiency**

Figure 5c shows the NSI as a function of the ratio of sample flow to sheath flow in the ionization region. The NSI firstly increased and then decreased with the increased ratio, with a peak value at a sample/sheath flow ratio of 0.3. This optimized ratio was independent of the total flow rates from 12 to 21 slpm. This ratio produced a turbulent flow in the chemical ionization region. Such flow dynamics facilitates a fast mixing of the reactants, and enhances the f(IE) of $H_2SO_4$ as well as the NSI at 97 m/z (Tanner and Eisele, 1995; Tanner et al., 1997).

Figure 5d-e shows the effects of voltages applied to the sample and sheath flow on NSI. The NSI first increased and then decreased as the increase of difference in voltage between the sample and sheath flow (figure 5d). At the voltage difference of 48 V, the peak NSI was achieved, and this voltage difference is selected. Figure 5e shows the NSI increased with the negative sheath voltage and then kept stable with sheath voltage < -70V. This shows that when it is negative charged, it is the voltage difference but not the exact voltage that affects the NSI. In our studies, we set the inlet and sheath voltages at -32 and -80 V, respectively. The cross interactions of sample/sheath flow and voltages on NSI were also evaluated (see Figure S3). The result shows that the highest NSI was achieved when the sample/sheath flow ratio was close to 0.3, independent of the voltages. The optimized f(IE) of the CIMS is achieved by the above selections of the flow ratio and electric field.

**5.4.4 Transmission efficiency**

The $N_2$ buffer flow rate controls the proportion of sample air in dry $N_2$ (refer to figure 1), thereby affecting amount of ion clusters in the sample air entering the mass detector. Figure 5f shows that the NSI increased with the decreased the buffer flow rate, as expected. However, a lower flow rate of $N_2$ buffer gas also allows more undesired neutral molecules and particles in sample air to enter the mass spectrometer (Berresheim et al., 2000). With these considerations, the flow rate of $N_2$ buffer gas was set as 440 sccm. To force the ions to the center of the pinhole, the voltage applied to before and on the pinhole was set at -70V and -40V, respectively.

**5.4.5 Noise minimization**

After optimization of CIMS's sensitivity, noise minimization is needed to reduce the signal that is not related to the ambient OH concentration. As discussed above, the noises for OH measurements include $H_2SO_4$ in ambient air and artificial OH produced by the ion source. These noises can be mitigated by adding a scavenger gas in the sheath flow to eliminate artificial OH and in sample flow to quantify ambient $H_2SO_4$ (Figure 1). Below we determine the optimal flow rates for the scavenge gas.

Figure 6 shows the HSO4- signal intensity as a function of propane flow rates in sheath flow, with $N_2$ gas as the sample air. When propane was not added, the artificial OH concentration from the [210]Po ion source was ~$3.5\times10^6$ molecules $cm^{-3}$, which is comparable to the typical OH

concentrations in ambient environments. When propane was added into the sheath gas, the artificial signals were reduced with the increasing propane flow and kept stable at ~$1 \times 10^6$ molecules cm$^{-3}$ when the flow rate was higher than 1 sccm. We thus set a flow rate of 2 sccm for propane in the sheath flow.

Figure 7 shows the removal efficiency (RE) of OH by propane added in the sample flow as a function of the propane flow rates. The OH radicals were produced by the calibration unit described in Section 4.2. The RE increased with the increased propane flow rate initially and levelled off at the flow rate > 1 sccm. We adopt the flow rate of propane of 2 sccm (~535 ppm), which led to ~98% removal efficiency for OH. As OH concentrations in this test are much higher than those in typical ambient air, the RE of the propane for ambient OH should be even large at the selected flow rate.

**6. Detection limit and uncertainty**

The detection limit can be calculated as follows,

$$DL = \frac{1}{C} \times \frac{n * \sigma}{\{NO_3^-\}} \tag{E6}$$

Where DL is the detection limit in $10^6$ molecule/cm$^3$, C is the calibration factor, and *n* is the ratio of signal to noise S/N. $\sigma$ represents the standard deviation of the signal intensity of $HSO_4^-$ at 97 m/z, and $\{NO_3^-\}$ represents the signal intensity of $NO_3^-$ at 64 m/z at the integration time t.

Figure 8 shows the concentrations of OH radicals and the corresponding detection limit (S/N=2, average time=6 minutes) in the laboratory. The detection limit was quite stable over the whole day and ranged from 0.08 to $0.20*10^6$ molecule cm$^{-3}$, with an average value of approximately $0.15 *10^6$ molecule cm$^{-3}$.

The uncertainty for the calibration factor (C) of OH measurements is dependent on the uncertainties of all the parameters involved in the calculation of the concentrations of OH radicals and the precision of the measurements of signal at 64 m/z and 97 m/z. The uncertainty was ~36% for *It* (see Figure 3), $\sigma_{H_2O}$~5% for $\sigma_{H_2O}$, <1% for $\phi_{H_2O}$ (Cantrell et al. 1997), and ~10% for the water concentration (Kukui et al., 2008). The precision of the measurements signal at 64 m/z and 97 m/z of the CIMS instrument ($2\sigma$) was 11% (for 6 min integration time). The overall uncertainty for the calibration factor was about 38%.

**7. Field deployment of CIMS**

In order to examine the performance of our CIMS in the ambient environment, we deployed the optimized instrument to an urban site of Hong Kong in April 2019 (Figure S4). The site was located on the 11[th] floor of a teaching building on the campus of The Hong Kong Polytechnic University (PolyU) and was surrounded by several busy roads. The sample inlet was positioned horizontally facing the south. Measurements were made with a time resolution of 10 seconds. A typical measurement sequence consisted of 3 minutes in the background mode and 3 minutes in the signal mode. Figure 9a shows the diurnal profile of OH concentrations (3-minute average) observed on April 25, 2019, and the solar radiation measured using UTA-LI200 at a time resolution of 1 minute. Figure 9b shows the measured signal intensities at 97 m/z at the signal mode and the background mode. The OH concentrations exhibited a clear diurnal profile with the highest value of $\sim 6 \times 10^6$ molecules cm$^{-3}$ at midday and the lowest level of $\sim 0.25 \times 10^6$ molecules cm$^{-3}$ at night. The OH concentrations were highly correlated to solar radiation, which was similar to previous studies (e.g. Rohrer and Berresheim, 2006; Tan et al., 2017). The 3-minute average OH concentrations were above the detection limits ($0.5$-$2 \times 10^6$ molecules cm$^{-3}$) most of the daytime, except during a cloudy period (08:00 to 10:00) (Figure 9a). This preliminary result demonstrated the capability of our CIMS for measuring ambient OH on a clear day in an urban environment. However, Figure 9b reveals that the contribution to instrument background from ambient $H_2SO_4$ was significant at the site, which raised the detection limit and measurement uncertainty (to 51%). The ambient $H_2SO_4$ concentration (10 sec average), which was estimated by the normalized background signal divided by the OH calibration factor, varied from $2.6 \times 10^5$ in before sunrise to $8 \times 10^6$ molecule/cm$^3$ in mid-day (figure not shown). Future work will make use of isotopically labelled $^{34}SO_2$ to eliminate $H_2SO_4$ interference (Eisele and Tanner, 1991).

**8. Summary and conclusions**

To measure the atmospheric OH radicals, we have developed the first chemical ionization mass spectrometry (CIMS) system in Asia. It is an indirect measurement technique that converts OH radicals to $HSO_4^-$ which is detected by the ion-assisted mass spectrometry method. In addition, the calibration system has been developed. A series of comparisons of different ion sources, scavenger gases, and primary ions detection have been conducted to optimize the performance of the CIMS for OH measurement. The sensitivity is dependent on the efficiencies of conversion reaction, ion conversion, and ion transmission which have been improved by optimizing the flow rates of a myriad of gases and voltages in various components. An initial field test has demonstrated the capacity of this instrument in measuring ambient OH in an urban site on clear days. The main findings on the key parameters are summarized below.

(1) $^{210}$Po has lower artificial OH interference compared to a corona ionizer, and it is adopted as the ion source.

(2) $C_3H_8$ is a better OH scavenger than $C_3F_6$ because of the high elimination efficiency and signal stability of $C_3H_8$.

(3) A set of procedures has been developed to optimize the flow rates of sample gas, sheath gas, and $N_2$ buffer gas, voltages on the sample inlet system and the concentration of $SO_2$ conversion gas with the aim to increase instrument's sensitivity and reduce noise.

(4) The CIMS instrument achieved a detection limit of $0.15 \times 10^6$ molecules $cm^{-3}$ and uncertainty of 38% (S/N=2) under laboratory conditions. In the field, the detection limit increased to about $0.15 \times 10^6$ molecules $cm^{-3}$ on clear days, with the overall accuracy of about 51%.

(5) Future work includes more field experiments in various environments and utilization of isotopically labelled $^{34}SO_2$ to eliminate the $H_2SO_4$ interference.

We note that the optimal values of instrument parameters may differ in different CIMS systems due to the different design and/or configurations, the test procedures and results from our study provide a useful reference to other researchers who wish to apply CIMS technique to measure atmospheric OH radicals.